# Wilkes subglacial basin ice sheet response to Southern Ocean warming during late Pleistocene interglacials

Ilaria Crotti [1,2] ✉, Aurélien Quiquet[3,4], Amaelle Landais [2], Barbara Stenni [1,5], David J. Wilson [6], Mirko Severi [5,7], Robert Mulvaney [8], Frank Wilhelms[9,10], Carlo Barbante [1,5] & Massimo Frezzotti [11]

The response of the East Antarctic Ice Sheet to past intervals of oceanic and atmospheric warming is still not well constrained but is critical for understanding both past and future sea-level change. Furthermore, the ice sheet in the Wilkes Subglacial Basin appears to have undergone thinning and ice discharge events during recent decades. Here we combine glaciological evidence on ice sheet elevation from the TALDICE ice core with offshore sedimentological records and ice sheet modelling experiments to reconstruct the ice dynamics in the Wilkes Subglacial Basin over the past 350,000 years. Our results indicate that the Wilkes Subglacial Basin experienced an extensive retreat 330,000 years ago and a more limited retreat 125,000 years ago. These changes coincide with warmer Southern Ocean temperatures and elevated global mean sea level during those interglacial periods, confirming the sensitivity of the Wilkes Subglacial Basin ice sheet to ocean warming and its potential role in sea-level change.

The growth and decay of polar ice sheets exert important controls on regional and global climate, while their future behaviour is a key uncertainty in predicting sea-level rise during and beyond this century[1]. Over the last decade, it has been observed that excess basal melting in Antarctica, arising from ocean heat supply, has increased the dynamic mass loss of grounded ice shelves bordering the Southern Ocean (SO)[2]. This observation has implications for future ice sheet stability, and also suggests that ocean warming may have played a role in controlling past ice sheet dynamics in Antarctica.

The vast marine-based West Antarctica Ice Sheet, which could contribute up to 4–5 m to Global Mean Sea Level (GMSL)[3], is well known for its past and future vulnerability to a warming climate[4,5]. In contrast, past stability of the much larger East Antarctic Ice Sheet (EAIS), which is characterized by a total potential contribution to

Global Mean Sea Level (GMSL) of 53 m[6], of which around one-third is marine-based ice, is still under debate[7–10]. The Wilkes Subglacial Basin, which contains 3 to 4 m sea-level equivalent[3], is characterized by a reverse-sloping bed with an elevation below sea level (Fig. 1). Ice that is grounded below sea level is vulnerable to intrusions of warm modified Circumpolar Deep Water (CDW) across the continental shelves into ice shelf cavities[3,11,12]. An initial grounding line retreat into deeper water may then lead to a marine ice sheet instability condition, which would be followed by increased ice discharge, inland thinning, and a rapid contribution to GMSL[13].

Satellite altimetry and images reveal that the Cook Glacier, which drains a large proportion of the Wilkes Subglacial Basin, has experienced thinning of 33 ± 12 cm/a (year) over the past 25 years[14], following the near-complete loss of the Cook West Ice Shelf between 1973 and

[1]Department of Environmental Sciences, Informatics and Statistics, Ca' Foscari University, Venice, Italy. [2]Laboratoire des Sciences du Climat et de l'Environnement LSCE/IPSL, CEA-CNRS-UVSQ, Université Paris-Saclay, Gif-sur-Yvette, France. [3]NumClim Solutions, Palaiseau, France. [4]Université Grenoble Alpes, CNRS, IRD, Grenoble INP, IGE, Grenoble, France. [5]Institute for Polar Sciences (ISP), CNR, Venice, Italy. [6]Institute of Earth and Planetary Sciences, University College London and Birkbeck, University of London, London, UK. [7]Department of Chemistry Ugo Schiff, University of Florence, Florence, Italy. [8]British Antarctic Survey, Cambridge, UK. [9]Alfred-Wegener-Institut Helmholtz-Zentrum für Polar-und Meeresforschung, Bremerhaven, Germany. [10]Geoscience Center, University of Göttingen, Göttingen, Germany. [11]Department of Science, Roma Tre University, Rome, Italy. ✉e-mail: ila.crotti@unive.it

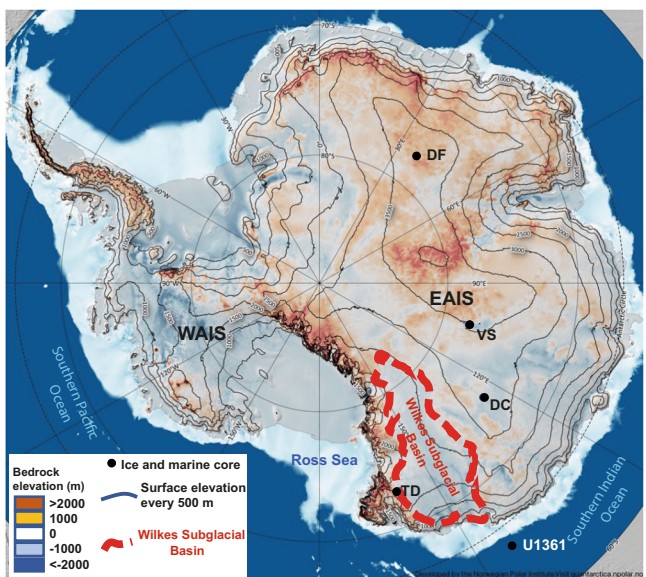

**Fig. 1 | Location of main Antarctic ice core drilling sites and the marine sediment core at Site U1361.** The map shows the subglacial bedrock elevation above sea level (colour scale, m) and the Antarctic ice sheet present-day surface elevation above sea level (contours, m)[6]. The ice core drilling sites of Talos Dome (TD), Dome C (DC), Vostok (VK), and Dome F (DF) and the marine sediment core at Site U1361 are indicated with blue dots. The studied area of the Wilkes Subglacial Basin is delimited by the red dashed contour. Map created using the Quantarctica GIS package[69] developed by the Norwegian Polar Institute and published under the Creative Commons Attribution 4.0 International License.

1989 that resulted from intense oceanic warming during the middle of the 20th century[15,16]. Model simulations suggest a modest sensitivity of the Wilkes Subglacial Basin ice sheet to oceanic warming[17] and margin retreat controlled by the presence of a coastal ice plug[18]. However, projected atmospheric and oceanic warming could soon lead to the crossing of tipping points in Antarctica, and hence to the destabilization of marine-based sectors of the ice sheet[19,20]. In light of this potential vulnerability, future predictions of the Wilkes Subglacial Basin ice dynamics should be refined by studying previous occurrences of instabilities during past warm climatic periods when temperatures were comparable to, or warmer than, modern conditions.

Here we explore the past ice dynamics of the Wilkes Subglacial Basin during the recent interglacial Marine Isotopic Stages (MIS) 5.5, 7.5, and 9.3 of the last 350 ka (thousand years ago). These warm periods can be considered similar, in terms of atmospheric warming and GMSL increases, to a range of near-future climate projections[19,21].

The Talos Dome Ice Core (TALDICE) (159°11′ E, 72°49′S, 2315 m a.s.l), which was drilled in a peripheral area of the East Antarctic Plateau (Fig. 1), has been expected to be sensitive to grounding line retreat in the Wilkes Subglacial Basin since the beginning of the project[9,22]. The recent extension of the TALDICE ice core chronology back to 343 ka (TALDICE-deep1) and the observation of a unique behaviour of the δD record during the interglacial periods[23] may therefore come helpful for deciphering the late Pleistocene dynamics of the peripheral EAIS.

Our approach is based on interrogation of the new isotopic data (δ¹⁸O and d-excess) from the TALDICE ice core at Talos Dome, in comparison to the EDC ice core record at Dome C, which is representative of East Antarctic Plateau conditions under the influence of the Southern Indian Ocean (Fig. 1). We compare our estimated elevation changes at Talos Dome with simulations of local ice thickness variations and Wilkes Subglacial Basin ice dynamics in experiments conducted with the GRISLI ice sheet model[24], in which we identify past instability events. To provide a comprehensive picture of the ice sheet behaviour in the Wilkes Subglacial Basin during past warm

interglacials, we also integrate our glaciological data and simulation results with late Pleistocene sedimentological and geochemical records from the International Ocean Discovery Program (IODP) Hole U1361A, offshore Wilkes Subglacial Basin[7] (Fig. 1).

Our analysis suggests that neither changes in air mass trajectories nor variations in sea-ice extent can explain the unique TALDICE δ¹⁸O signal recorded during late MIS 5.5, MIS 7.5, and MIS 9.3. Instead, we propose that the interglacial anomalies in the isotopic record were produced by lowering of the site elevation at Talos Dome due to ice loss and inland retreat of the Wilkes Subglacial Basin grounding line in response to the intrusion of warmer ocean waters. The GRISLI ice sheet simulation that best fits with our elevation data suggests a 10% reduction of the Wilkes Subglacial Basin ice volume during MIS 5.5, and a loss of up to 25% during MIS 9.3. Hence, these findings depict a highly dynamic ice sheet in the Wilkes Subglacial Basin and provide insights into the future response of the EAIS in a warmer world.

## Results

### TALDICE isotopic records over past interglacial periods

The EDC and TALDICE water isotopic records are both influenced by precipitation originating mainly from the Southern Indian Ocean[25]. The site of EDC on the East Antarctic Plateau is believed to be predominantly representative of past climatic variations at a hemispheric scale[26], while the TALDICE isotopic record is also sensitive to localised sea-ice extent in the Ross Sea[27,28] and local elevation changes[9,26,29]. During the current and Last Interglacial (LIG) periods, TALDICE and the plateau ice cores (EDC, Vostok, Dome F) share common isotopic maxima (δ¹⁸O and δD), between 12 and 9 ka in the Holocene and at 128 ka during the LIG[30]. The LIG isotopic maxima indicate Antarctic atmospheric temperatures 2–4.5 °C warmer than the Holocene[31–33], probably arising from the operation of the bipolar seesaw mechanism during deglaciation of the Northern Hemisphere ice sheets[30,34,35]. However, during the late stage of MIS 5.5, the TALDICE isotopic record shows a δ¹⁸O peak at 117 ka followed by an abrupt decrease towards the glacial inception, and such features are not recorded in the cores from the plateau[30].

Here we compare our new TALDICE δ¹⁸O data (measured at 5 cm resolution) from MIS 5.5[30], MIS 7.5, and MIS 9.3 with the published EDC δ¹⁸O record[36,37], which is representative of the Antarctic Plateau conditions, in order to investigate TALDICE isotopic patterns during previous late Pleistocene interglacials (Methods). Interestingly, the TALDICE δ¹⁸O record from MIS 9.3 exhibits a double-peak shape that is similar to the one observed for MIS 5.5, which is not seen in the EDC record (Fig. 2c). Both cores display a common interglacial isotopic peak at 335 ka, but from 331 ka the TALDICE signal diverges from the EDC signal, showing a sustained increase of about 1.5‰ until 326 ka followed by a steep decrease, while the EDC record declines gradually across this entire interval. During MIS 7.5, there is also a divergence, but of smaller magnitude, between the two records (Fig. 2c). The TALDICE δ¹⁸O record exhibits a second late and muted rise of ~0.8‰ between 240 ka and 237 ka, rather than the well-defined late peak observed for MIS 5.5 and MIS 9.3, while the EDC δ¹⁸O record decreases towards the glacial inception. In the case of MIS 7.5, the EDC and TALDICE isotopic records also diverge before their interglacial maxima, at around 245 ka. However, this offset is probably due to the uncertainties associated with the TALDICE deep1 chronology driven by the low resolution of the δ¹⁸O_atm record[23] and should not be interpreted as a physical signal. A new set of high-resolution δ¹⁸O_atm measurements would be needed to further investigate this point.

We explore three main hypotheses that could explain the discrepancies between the EDC and TALDICE water isotopic records, namely (i) differences in moisture sources, (ii) changes in sea-ice extent in the Ross Sea, and (iii) a local decrease of elevation at Talos Dome.

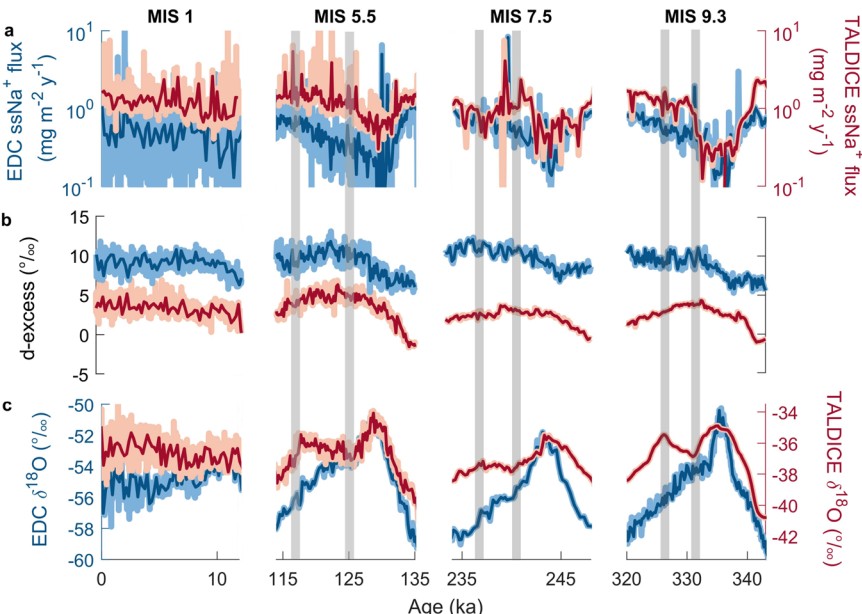

**Fig. 2 | Compilation of Talos Dome ice core (TALDICE) and Epica Dome C ice core (EDC) sea-salt sodium fluxes, d-excess, and δ18O records over the last four interglacial periods.** For all the proxies, the resampled records at 200 year intervals (blue curves for EDC and red curves for TALDICE) are superimposed on the raw signals (light blue curves for EDC and pink curves for TALDICE). **a** EDC[36, 37] and TALDICE[28, 44] ssNa+ fluxes on logarithmic scale. TALDICE ssNa+ fluxes for Marine Isotopic Stage (MIS) 7.5 and 9.3 are from this study. **b** EDC[41] and TALDICE[28] d-excess records. TALDICE d-excess records for MIS 5.5, 7.5, and 9.3 are from this study. **c** EDC[36,37,41] and TALDICE[30,34] δ18O records. TALDICE δ18O data for MIS 7.5 and 9.3 are from this study. The anomalies in the TALDICE δ18O record are identified by grey shaded bars (marking the start and end of the anomalous increase in δ18O values).

Differences in moisture sources between EDC and TALDICE can be explored using the deuterium excess (*d*-excess = δD − 8·δ18O) thanks to the new TALDICE *d*-excess profile (5 cm resolution) for MIS 5.5, 7.5, and 9.3. This second-order parameter is an indicator of climate conditions in the vapour source regions, and is therefore sensitive to changes in the provenance areas or changes in air mass trajectories towards the sites[38–40]. Our comparison shows no clear differences between the TALDICE and EDC[41] *d*-excess profiles, which exhibit consistent patterns during interglacial stages (Fig. 2b), as already observed for shorter warming events[42]. In particular, the increases in *d*-excess for TALDICE and EDC during MIS 5.5, MIS 7.5, and MIS 9.3 occur over longer time intervals than the δ18O increase, and the maxima in *d*-excess are reached in the second half of the interglacial period[41] but before the second δ18O anomaly in the TALDICE record. In summary, the absence of peculiarities in the *d*-excess records during the intervals with δ18O anomalies at TALDICE suggests that there are no significant variations of moisture sources[40,43] that could explain the TALDICE isotopic behaviour.

The hypothesis of a change in sea-ice extent can be addressed using published sea-salt sodium (ssNa+) fluxes in TALDICE for MIS 1[28] and MIS 5.5[44], and new data for MIS 7.5 and MIS 9.3, as a proxy for sea-ice coverage in the western sector of the Ross Sea and in the southern Indian Ocean facing the Wilkes Subglacial Basin[28]. The high-resolution (7-8 cm) TALDICE ssNa+ flux record for MIS 5.5[44], MIS 7.5, and MIS 9.3 agrees well with the EDC record[36,37] and no site-specific differences are identifiable (Fig. 2a). Note that the peak in the TALDICE ssNa+ record during MIS 7.5 at 240 ka is not interpreted as being climate-driven, but rather as an artifact of chemical weathering processes in the deep ice layers[45,46]. The overall coherence between the EDC and TALDICE ssNa+ records indicates that sea-ice variations in the Ross Sea or southern Indian Ocean cannot explain the double-peak shape of the δ18O record at Talos Dome, and hence the second hypothesis can also be discounted.

In summary, our multi-proxy comparison (Fig. 2) indicates that neither changes in air mass trajectories nor in sea-ice extent can explain the unique interglacial δ18O excursions recorded at Talos

Dome. In the following sections, we therefore interpret the isotopic anomalies as an indication of elevation changes at this site, and estimate the magnitude of changes required to explain the anomalies.

## Interglacial elevation decrease at Talos Dome from δ18O records

The relationship between δ18O values and ice sheet elevation has recently been investigated to reconstruct EAIS dynamics during the Last Glacial Maximum (LGM) and LIG[9,29,47]. To estimate the imprint of elevation changes at Talos Dome during the LIG, Sutter et al.[9] applied a δ18O-elevation relationship of −0.53‰/100 m, based on multiplying the present day lapse rate of −0.8 °C/100 m by the local δ18O isotope-temperature relationship of 0.66‰/°C estimated from the atmospheric general circulation model ECHAM5-wiso equipped with an isotope module[47]. In contrast, Goursaud et al.[29] obtained a relationship of −0.93‰/100 m for the LIG, based on the simulated isotopic response to idealised changes in Antarctic ice sheet elevation in the isotope-enabled coupled ocean-atmosphere-sea-ice general circulation model HadCM3[48]. As a third approach, we directly compute the isotope-elevation relationship from present-day snow-pit δ18O data collected along the traverse from GV7 site to Talos Dome[49], which takes into account the provenance of air masses reaching Talos Dome (see details in "Methods"). This method leads to an estimate of the modern isotopic lapse rate for Talos Dome of −1.35 ‰/100 m.

We use the three different δ18O lapse rate estimates for TALDICE to calculate the elevation changes required to explain the observed anomalous increase of the isotopic signal (Δδ18O) (see "Methods") for MIS 5.5 (+1.68‰), MIS 7.5 (+0.68‰), and MIS 9.3 (+1.42‰) (Table 1). The less negative isotopic values during the late stages of the interglacials are connected to an increase in temperature at Talos Dome caused by elevation reduction at the site. The periods of MIS 5.5 and MIS 9.3 were apparently subjected to the largest elevation decrease, with a similar magnitude for the two periods, while only around half of this elevation decrease is invoked to explain MIS 7.5 signal. Applying the isotopic lapse rates of −1.35‰/m[49] and −0.93‰/m[29], we obtain a

**Table 1 | Isotopic anomalies and elevation changes calculated and modelled for Talos Dome during interglacial MIS 5.5, 7.5, and 9.3**

| | | MIS 5.5 | MIS 7.5 | MIS 9.3 |
|---|---|---|---|---|
| Time interval of δ¹⁸O anomaly (ka) at TALDICE | | 117–127 | 237–240 | 326–331 |
| δ¹⁸O max and δ¹⁸O min (‰) | | (−35.45)–(−37.13) | (−37.19)–(−37.87) | (−35.46)–(−36.89) |
| Δδ¹⁸O (‰) | | +1.68 | +0.68 | +1.42 |
| **Lapse rate (‰/100m)** | | **Elevation changes (m)** | | |
| −0.53 (Sutter et al. 2020) | | −317 | −128 | −268 |
| −0.93 (Goursaud et al. 2020) | | −180 | −73 | −153 |
| −1.35 (Magand et al. 2004) | | −124 | −50 | −105 |
| | **Ocean forcing** | **MIS 5.5** | **MIS 7.5** | **MIS 9.3** |
| | | **GRISLI elevation changes at Talos Dome (m)** | | |
| Time interval for max elevation anomaly at TALDICE (ka) | | 115–128 | 233–241 | 321–332 |
| DS | Quiquet et al. (2018) | −123 | −103 | −757 |
| DS-5 | Quiquet et al. (2018) +5% | −114 | −104 | −754 |
| DS-10 | Quiquet et al. (2018) +10% | −146 | −105 | −749 |
| GS | Quiquet et al. (2018) | −126 | −85 | −134 |
| GS-5 | Quiquet et al. (2018) +5% | −116 | −89 | −473 |
| GS-10 | Quiquet et al. (2018) +10% | −152 | −103 | −754 |

Isotopic anomalies are calculated from the δ¹⁸O record resampled at 200 year intervals. Elevation changes are calculated from the isotopic anomalies using the lapse rate estimates of Sutter et al.[9], Goursaud et al.[29], and Magand et al.[49]. Deglaciated Start (DS) and Glacial Start (GS) elevation changes are modelled for 6 sensitivity tests performed with the Grenoble Ice Sheet and Land Ice (GRISLI) ice sheet model, varying the Antarctic ice sheet initial conditions and the Southern Ocean (SO) temperature forcing. DS simulations are initialized with Antarctic interglacial initial conditions at 400 ka and GS simulations are initialized with Antarctic glacial initial conditions at 400 ka. Elevation variations at Talos Dome are calculated for the interglacial time intervals when the simulated GRISLI ice thickness variations are maximized.

decrease in elevation of around −100 to −200 m during MIS 5.5 and MIS 9.3. Using the lapse rate of −0.53‰/m[9] would imply a larger elevation reduction (~−300 m) for those intervals. Hence, while subject to uncertainty from the choice of lapse rate, we conclude from the TALDICE isotopic record that the Talos Dome site was subjected to an elevation decrease on the order of −100 to −300 m during MIS 5.5 and MIS 9.3 (Table 1). We further note that, due to signal smoothing at the centennial resolution of the record in the deep portion of the ice core[23], the Δδ¹⁸O anomaly during MIS 9.3, and consequently the ice thickness reduction inferred for this interglacial, could be underestimated.

## Sensitivity tests with the GRISLI ice sheet model

To further explore possible elevation changes at Talos Dome, as well as Wilkes Subglacial Basin grounding line displacements, we perform numerical experiments of the Antarctic ice sheet dynamics over the last 400 ka with the GRISLI ice sheet model[24], using the same setup as Quiquet et al.[24]. The model is forced, for the past 400 ka, by near-surface air temperatures over Antarctica deduced from the EDC δD record[24,31], which represents a boundary condition at the surface of the ice sheet for thermomechanical coupling and drives local changes in precipitation and surface mass balance. On the other hand, the oceanic forcing is produced through modification of the present-day sub-shelf melt rate by a palaeo-oceanic index based on the ODP 980 benthic temperatures in the North Atlantic Ocean[24,50], which represent North Atlantic Deep Water (NADW) temperatures. The NADW is produced by deep convection of dense waters at high northern latitudes, then flows southward in the deep Atlantic Ocean, and upwells in the SO. Due to the lack of a suitable SO proxy record for sub-surface oceanic temperatures spanning the last 400 ka, this estimate of upwelled NADW temperature is a reasonable indicator for the oceanic heat available in Antarctic coastal regions that affects the sub-shelf melting rate at the grounding line. We have also performed additional sensitivity tests using two other estimates of the sub-surface oceanic conditions in the SO (Supplementary Fig. 2) and those results are presented in the Supplementary Information. This simplified approach does not account for the influence of local sea-ice changes[27] or freshwater capping[35,51] on sub-surface temperatures

around Antarctica, or for variability in Circumpolar Deep Water upwelling or cross-shelf transport[52]. Considering the simplified SO conditions employed, we modify the original oceanic forcing to test the response of the Wilkes Subglacial Basin ice sheet to variations of the oceanic conditions.

We present 6 sensitivity experiments, varying (i) the initial state of the Antarctic ice sheet at 400 ka, and (ii) the sub-shelf melting rate. The GRISLI experiments labelled DS (Deglaciated Start) adopt an initial Antarctic interglacial ice sheet state, while the GS (Glacial Start) experiments use a Last Glacial Maximum (21 ka) initial state[24]. The initial ice sheet GS state is simulated as in Quiquet et al.[24], while the DS state has been defined for this study and is obtained after 10 ka simulation under present-day climate forcing (1976–2016) and doubling the original sub-shelf melting[24], starting from the Interglacial Start (IS) represented by the present-day Antarctic ice sheet state. The DS state is considered more likely to reproduce Antarctic ice sheet conditions during MIS 11 (~400 ka) in comparison to the IS state. Simulations have also been performed applying a IS state and those results are presented in Supplementary Table 1 and Supplementary Figs. 3 and 4. For both DS and GS initial states, we present three GRISLI simulations forced by different ocean sub-surface conditions (see "Methods"): (i) the sub-shelf melting rate signal applied by Quiquet et al.[24] and derived from the ODP 980 benthic record[50], (ii) the signal defined by Quiquet et al.[24] increased by 5%, and (iii) the signal defined by Quiquet et al.[24] increased by 10% over the past 400 ka (Table 1). The GRISLI experimental results include elevation changes at Talos Dome, ice volume variations in the Wilkes Subglacial Basin, grounding line displacements, and the contribution of grounded ice above flotation to GMSL.

The ice thickness results from the GS and DS sensitivity experiments are compared to ice thickness variations estimated from the isotopic record at Talos Dome (Table 1 and Fig. 3), with simulation GS-5 (i.e. Glacial Start and +5% oceanic forcing) showing the best agreement with the elevation changes deduced from the ice core record. For MIS 5.5, simulation GS-5 predicts an elevation decrease at Talos Dome on the order of 100 m (Fig. 3c, Table 1, Supplementary Fig. 5a), an average of ~100 km grounding line retreat into the Wilkes Subglacial Basin between 133 and 115 ka (Fig. 4a), and loss of 10% of its ice volume (Fig. 3d).

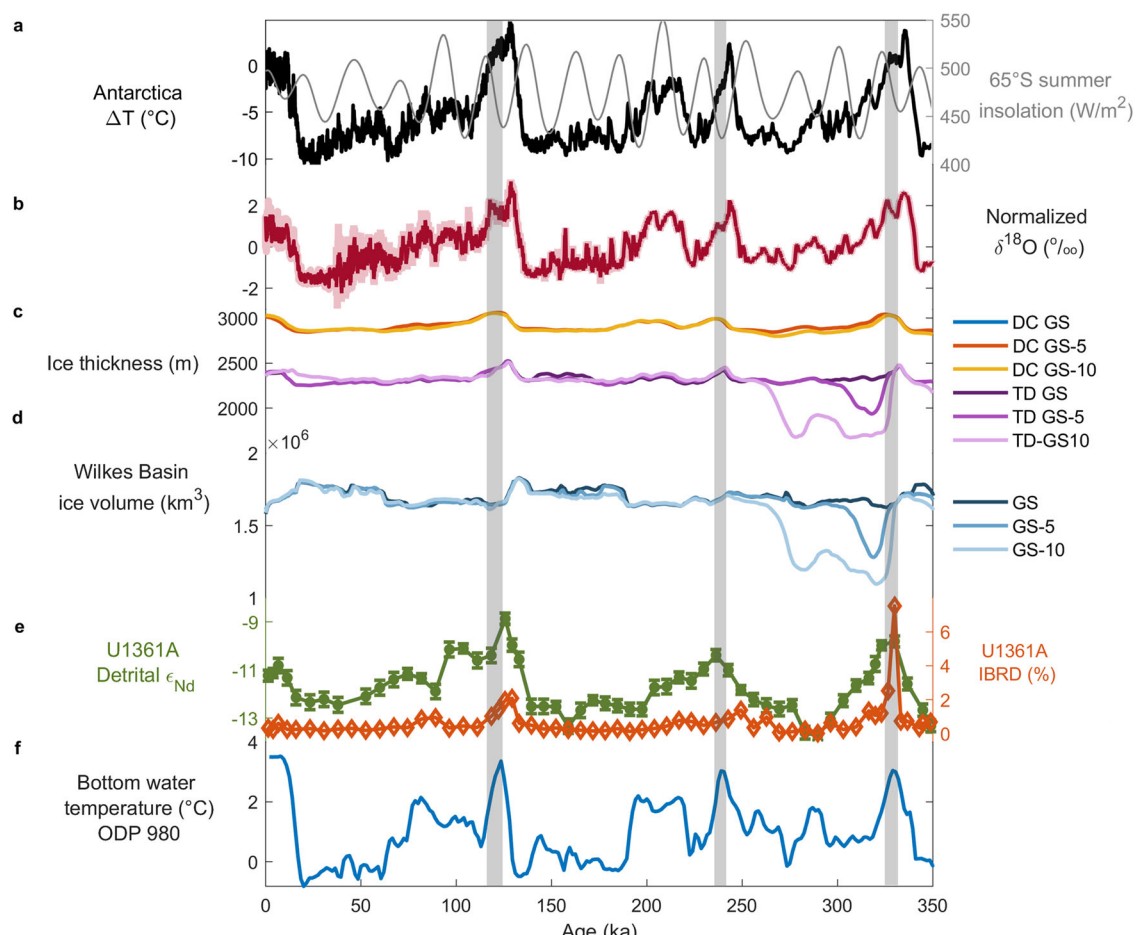

**Fig. 3 | Comparison of the Talos Dome ice core (TALDICE) isotopic record to Grenoble Ice Sheet and Land Ice (GRISLI) simulations, sedimentological data from sediment core U1361A, and Antarctic atmospheric and oceanic temperature records since 350 ka. a** Antarctic ice core temperature difference (ΔT, difference from mean values of the last millennium) derived from δD at Epica Dome C (EDC)[31] plotted on the Antarctic Ice Core Chronology 2012 (AICC2012)[66] and 65°S summer insolation (grey curve)[70]. **b** TALDICE normalized δ[18]O record (data centred and scaled to have mean 0 and standard deviation 1) (measured, pink; resampled at 200 year intervals, red). Holocene data are from Stenni et al.[34] and Marine Isotopic Stage (MIS) 5.5 data are from Masson-Delmotte et al.[30]. The MIS 7.5 and MIS 9.3 isotopic data are from this study. **c** Talos Dome (TD) and Dome C (DC) elevation from Glacial Start (GS) simulations, based on the original oceanic forcing used by Quiquet et al.[24], and the original forcing increased by 5% and by 10%. **d** Wilkes Subglacial Basin ice volume evolution from GS simulations. **e** Core U1361A Nd isotope record (plotted as $\epsilon_{Nd}$; green curve with dots with their respective standard deviation) and iceberg-rafted debris (IBRD) % (brown curve with dots)[7] on AICC2012 age scale (see Methods). **f** North Atlantic bottom water temperatures (°C) derived from the Ocean Drilling Program (ODP) 980 δ[18]O benthic foraminifera record[50] used to derive the GRISLI oceanic forcing[24]. Grey bars highlight the intervals with a unique isotopic signal in the TALDICE ice core compared to the EDC record.

The main grounding line retreat during MIS 5.5 is seen earlier in the model (128 ka) than is suggested by the TALDICE isotopic signal (117–127 ka) (Table 1). However, we do not expect a perfect synchronicity between the two-time series because of uncertainties in their relative timescales and in the ice sheet time response. Moreover, the elevation change at Talos Dome may not be completely connected to the grounding line retreat, and could also be influenced by a change in accumulation rate during interglacials at the site.

For MIS 7.5, GS-5 predicts no grounding line displacement and only a very limited reduction in ice thickness at Talos Dome, in agreement with the TALDICE isotopic record (Fig. 3b, c, Table 1). For MIS 9.3, GS-5 simulates a more dynamic ice sheet response in the Wilkes Subglacial Basin, with an average of ~330 km grounding line retreat between 339 ka and 318 ka (Fig. 4b) leading to the loss of ~25% of its ice volume (Fig. 3d). The modelled elevation at Talos Dome exhibits an abrupt decrease of ~470 m between 332 and 321 ka (Fig. 3c, Supplementary Fig. 5b), synchronous with the TALDICE isotopic anomaly (Fig. 3b). However, the ice thickness variations simulated at Talos Dome during MIS 9.3 are much greater than those calculated from the isotopic record (~100–250 m) (Table 1). The

lower temporal resolution (~200 years/5 cm) of the TALDICE record below 1530 m depth probably leads to smoothing of the isotopic signal[23], which could be responsible for a muted Δδ[18]O signal during MIS 9.3. Overall, the modelled contribution from grounded ice above flotation in the Wilkes Subglacial Basin to GMSL increase is estimated to be +0.5 m during MIS 5.5 and +0.9 m during MIS 9.3 for the GS-5 experiment. The former value is supported by the changes in the TALDICE ice core isotopic record, while the latter value may be an upper estimate, depending on the extent of signal smoothing during MIS 9.3.

The results of the DS experiments depict a highly unstable ice sheet in the Wilkes Subglacial Basin during MIS 9.3, which is inconsistent with the elevation changes deduced from the TALDICE δ[18]O record and with evidence from previous studies[8,10]. A ~700 m elevation reduction at Talos Dome is simulated over MIS 9.3 and a significant grounding line retreat of the Wilkes Subglacial Basin ice sheet is modelled for all the DS experiments (Table 1, Supplementary Table 1). On the other hand, the DS simulations depict a less dynamic Wilkes Subglacial Basin ice sheet for MIS 5.5. An increase of 10% in the oceanic warming forcing seems to represent a tipping point that triggers

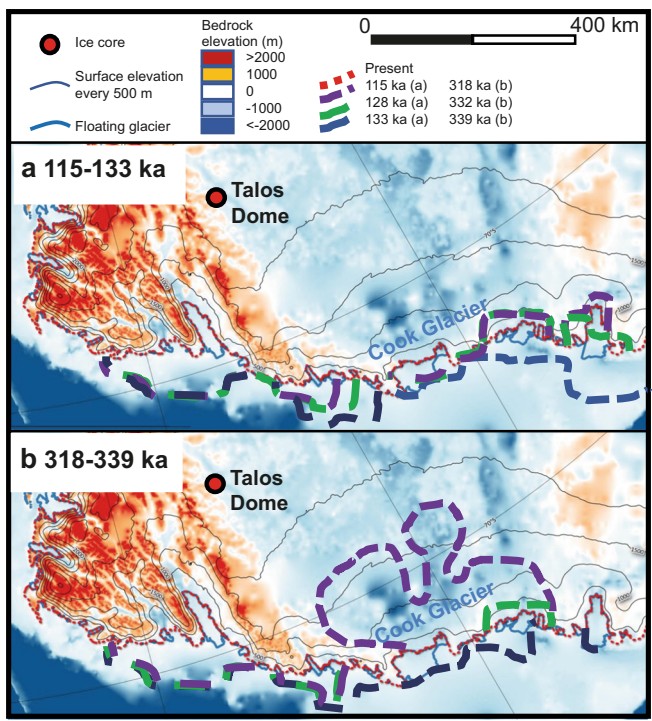

**Fig. 4 | Interglacial grounding line retreat of the Wilkes Subglacial Basin ice sheet simulated by the Grenoble Ice Sheet and Land Ice (GRISLI) GS-5 experiment at three different time intervals at 40 km resolution. a** Grounding line displacement during Marine Isotopic Stage (MIS) 5.5 at 133 ka (blue dashed line), 128 ka (green dashed line) and 115 ka (purple dashed line). **b** Grounding line displacement during MIS 9.3 at 339 ka (blue dashed line), 332 ka (green dashed line) and 318 ka (purple dashed line). The map also shows the subglacial bedrock elevation above sea level (colour shading, m), the Antarctic ice sheet present-day surface elevation above sea level (contours, m), and the present-day grounding line position[6] (red dashed line). Map created using the Quantarctica GIS package[69] developed by the Norwegian Polar Institute and published under the Creative Commons Attribution 4.0 International License.

deglaciation of the Wilkes Subglacial Basin ice sheet in both DS and GS experiments during MIS 9.3 (Table 1, Supplementary Table 1). We suggest that this tipping point was not passed during those intervals (Fig. 3).

**Comparison to the U1361A marine sediment core record**
The Antarctic ice sheet loses the majority of its mass via iceberg calving and sub-ice-shelf melting[53]. Past retreat of the ice shelves and ice sheet in the vicinity of the Wilkes Subglacial Basin is therefore expected to have left geochemical and sedimentological signatures in the nearby marine sediments. The sediment core U1361A (64.41°S, 143.89°E, 3,454 m water depth)[54], recovered from the continental rise adjacent to the Wilkes Subglacial Basin (Fig. 1), provides a near-continuous archive of Pliocene to Pleistocene variability of this marine EAIS margin[7,55,56]. Specific events of iceberg-rafted debris (IBRD) discharge can indicate dynamic ice loss, while Nd isotopes measured on detrital sediments ($\varepsilon_{Nd}$) provide a provenance indicator for changes in subglacial erosion and transport that could reflect ice sheet retreat[7,55,56] (Fig. 4). To enable a direct comparison of the TALDICE isotopic record and GRISLI simulations with the U1361A record, we refined the original U1361A chronostratigraphy to achieve consistency with the AICC2012 framework (see Methods and Supplementary Fig. 1).

The prominent IBRD and Nd isotope peaks during MIS 5.5 and MIS 9.3 (Fig. 3e) are interpreted as evidence of ice loss from the margins of the Wilkes Subglacial Basin[7]. Interestingly, the IBRD peaks, which record transient events of ice discharge and iceberg

calving, and the Nd isotope maxima, representing inland erosion, coincide with the onset of the TALDICE $\delta^{18}O$ anomalies at ~125 ka during MIS 5.5 and at ~330 ka during MIS 9.3 (Fig. 3b, e). In addition, the duration of the TALDICE isotopic anomalies are consistent with the sustained high Nd isotope values during the later stages of MIS 5.5 and MIS 9.3, suggesting a prolonged interval of inland glacial erosion. Furthermore, the highest IBRD content in the U1361A record (~7.5%) occurred at ~330 ka, indicating a significant ice loss event and potentially a more dynamic Wilkes Subglacial Basin ice sheet during MIS 9.3 in comparison to MIS 5.5, for which the IBRD peak was less pronounced (~2%) (Fig. 3e). A more dynamic behaviour during MIS 9.3 than MIS 5.5 is also supported by the GRISLI modelling results (Fig. 3c, d).

In contrast, the interglacial periods of MIS 1 and MIS 7.5 were characterized by only minor IBRD occurrences and more muted Nd isotope maxima in core U1361A (Fig. 3e), indicating a more stable ice sheet. Similarly, the TALDICE isotopic anomaly was less pronounced at these times (Fig. 3b), and the GRISLI simulations also indicate a dampened response in the Wilkes Subglacial Basin during those interglacials (Table 1, Fig. 3c, d). As such, evidence from these independent datasets and approaches appears to converge on a consistent picture of the differential response of the ice sheet in the Wilkes Subglacial Basin to the subtly different climate forcing of individual late Pleistocene interglacials.

## Discussion
By combining the TALDICE $\delta^{18}O$ record with GRISLI ice sheet model outputs and records from marine sediment core U1361A, we propose that the TALDICE isotopic anomalies during late Pleistocene interglacials reflect a reduction in elevation at Talos Dome arising from accumulation rate changes, ice loss, and grounding line retreat in the Wilkes Subglacial Basin. Our results from the GRISLI GS-5 simulation suggest that even a small increase of the SO sub-shelf melting rate of 5%, which we attribute to the intrusion of warm water at the grounding line depth, was enough to trigger significant margin retreat (although not complete collapse) of the ice sheet in the Wilkes Subglacial Basin during the warmest late Pleistocene interglacials. Notably, our simulated Talos Dome elevation changes during MIS 5.5 are consistent with recent studies[9,10], while the GRISLI GS-5 simulation depicts a larger retreat at the margin of the Wilkes Subglacial Basin which is in closer accordance with the U1361A record[7].

During interglacial periods, the combination of increasing insolation and decreasing ice sheet volume leads to the steepening of the meridional temperature gradient, which strengthens and shifts the Southern Westerly Winds poleward, promoting upwelling in the Southern Ocean[42,57]. In addition, the deglacial weakening of the Atlantic Meridional Overturning Circulation may have led to warmer Southern Ocean temperatures through the bipolar seesaw mechanism, with early interglacial atmospheric warming also attributable to $CO_2$ escape from deep waters[41,58,59]. The more southerly and warmer Antarctic Circumpolar Current could have induced a dynamic response in the EAIS during MIS 5.5[60,61] and a comparable scenario can also be anticipated for MIS 9.3. The distinctive late interglacial $\delta^{18}O$ peaks in the TALDICE record during MIS 5.5 and MIS 9.3 occurred ~9–11 ka after the interglacial isotopic maxima and are synchronous with the local summer insolation maxima at 65°S (Fig. 3a), which supports the hypothesis that local insolation could play a role, together with ocean temperature and upwelling, in modulating the ice sheet margin dynamics in the Wilkes Subglacial Basin[62,63].

Interestingly, a new late Pleistocene Subantarctic sea surface temperature reconstruction from the Indian sector of the SO (core DCR-1PC) shows that both MIS 5.5 and MIS 9.3 were characterized by a double warming phase[58]. The first warming phase corresponded to the interglacial optimum, which was followed by a cooling phase attributed to feedbacks from the Antarctic Ice Sheet, and then a second late

**Table 2 | Comparison of temporal resolution for Talos Dome ice core (TALDICE) and Epica Dome C ice core (EDC) $\delta^{18}O$ samples during interglacial periods**

| | | MIS 1 (0–20 ka) | MIS 5.5 (110–140 ka) | MIS 7.5 (234–252 ka) | MIS 9.3 (320–345 ka) |
|---|---|---|---|---|---|
| TALDICE | Sample length (cm) | 100 cm | 5 cm | 5 cm | 5 cm |
| | Resolution (years/5 cm) | – | 32 | 95 | 192 |
| | Resolution (years/m) | 24 | 630 | 1891 | 3843 |
| EDC | Sample length (cm) | 55 cm | 55 cm | 55 cm | 11 cm |
| | Resolution (years/sample length - cm-) | 22 | 53 | 116 | 33 |
| | Resolution (years/m) | 40 | 96 | 212 | 301 |

We calculated the sample resolution based on the original sample length (5 cm for TALDICE, 55 cm or 11 cm for EDC). We also calculated the average temporal resolution for each metre of core.

warming phase coincident with increasing local summer insolation, before the glacial inception[58] (Fig. 3f). Over time, the two long warming phases probably contributed to the destabilization of the ice shelves and outlet glaciers of the Wilkes Subglacial Basin, leading to inland retreat of the grounding line. Our evidence also points to a Wilkes Subglacial Basin ice sheet susceptible to local SO warming and/or CDW intrusion during MIS 9.3[58], suggesting a relatively unstable ice sheet state following the proposed retreat of about 700 km inland (from the present day position) during MIS 11[7,8].

Finally, sea-level reconstructions for MIS 5.5 and MIS 9.3 are also consistent with our reconstruction of the Wilkes Subglacial Basin ice dynamics during the late Pleistocene. Data from corals and other sea-level proxies indicate a late peak in GMSL during MIS 5.5 of ~6–9 m above present at ~119 ka[21,64]. For MIS 9, a stacked GMSL reconstruction based on various proxies indicates a sea-level highstand of ~9 m above present[65]. The GS-5 experiment indicates that the grounding line in the Wilkes Subglacial Basin may have retreated by several hundred kilometres during those interglacials, leading to contributions of approximately +0.5 m and +0.9 m to the GMSL increases during MIS 5.5 and MIS 9.3, respectively. These instabilities of the Wilkes Subglacial Basin ice sheet appear to have been driven mainly by increases in SO temperatures and/or CDW intrusion during past interglacials.

Despite the qualitative agreement between the ice core data, marine sediment core proxies, and the ice sheet modelling results, it will be essential to better assess the quantitative differences between the data and models, as well as the relative roles of atmospheric and oceanic warming in triggering ice sheet instabilities[17], in order to improve our understanding of EAIS dynamics during past interglacial intervals and on a future warming Earth.

## Methods

### TALDICE $\delta^{18}O$ and $\delta D$ records

In this study we focus on the oxygen and hydrogen isotopic composition ($\delta^{18}O$ and $\delta D$) and $d$-excess ($d = \delta D − 8 \times \delta^{18}O$) of new and published profiles measured at high resolution (5 cm) in the TALDICE ice core during interglacials. Here we present the published $\delta^{18}O$ bag resolution profile (1 m) for the Holocene[34], the published profile for MIS 5.5[30], and new 5 cm resolution profiles for MIS 7.5 and MIS 9.3. For the $d$-excess, we show the published bag resolution signal (1 m) for MIS 1[28] and the new 5 cm resolution signal for the oldest interglacials. We plot data from MIS 1 (0–20 ka) and MIS 5.5 (~115–132 ka) on the AICC2012 age scale[66]. The new data for $\delta^{18}O$ and $d$-excess during MIS 7.5 (~240–246 ka) and MIS 9.3 (~324–339 ka) are plotted on the TALDICE-deep1 age scale[23].

The 5 cm samples were analysed in Italy (University of Venice) and France (LSCE) using the Cavity Ring Down Spectroscopy (CRDS) technique. Analyses were performed using a Picarro isotope water analyser (L2130-i version for both laboratories). The data were calibrated using a three-point linear calibration with three lab-standards that were themselves calibrated versus Standard Mean Ocean Water

(SMOW). Intercomparisons between the two laboratories have been performed over the analysis period. The average precision for the $\delta^{18}O$ and $\delta D$ measurements is 0.1 and 0.7‰, respectively.

In this work we compare the $\delta^{18}O$ and $d$-excess records from the TALDICE and EDC cores[41], which are characterized by different temporal resolution, mostly due to differences in the sampling interval, snow accumulation, and thinning function at different depths (Table 2).

### Identification of change-points and calculation of anomalies

To identify differences between the TALDICE and EDC isotopic records, we assess changes in slope and intercept in the data sets. We first normalized both records (data centred and scaled to have mean 0 and standard deviation 1) and resampled them at the same time step of 200 year. We then searched for changes using the MATLAB task *find change points* with a maximum of 3 or 4 change points. The software identifies points with changes in slope (derivative) and intercept (Fig. 5). Isotopic anomalies ($\Delta\delta^{18}O$) were then calculated as the differences between values in the resampled record between the two change points (Table 1).

### Lapse rate calculation for TALDICE

In the modern day, Talos Dome receives 50% of its total precipitation from the west (Southern Indian Ocean), 30% from the east (Ross Sea and Southern Pacific Ocean), and ~15% from the interior[25]. In contrast, at EDC the modern precipitation originates mainly from the western Southern Indian Ocean (85%), with a small amount coming from the east via the Ross Sea and Transantarctic Mountains (15%).

The isotopic lapse rate expresses the variation in the oxygen isotopic composition ($\delta^{18}O$) in permil (‰) for every 100 m change in altitude. Here we calculate the isotopic lapse rate for TALDICE using the ITASE traverse dataset[49]. We take into account the $\delta^{18}O$ values of snow between the GV7 and Talos Dome sites sampled every 5 km and the altitude profile from the GV7 site to Talos Dome. This approach seems reasonable, given that the air masses travelling to Talos Dome mainly originate from the Southern Indian Ocean at 60°S and likely follow a similar path to the traverse once they reach Antarctica[25]. The calculated isotopic lapse rate is −1.35‰/100 m.

### TALDICE and EDC ssNa$^+$ fluxes

In this study we present the sea-salt sodium (ssNa$^+$) flux measured in the TALDICE ice core during MIS 1 (1 m resolution)[28], MIS 5.5 (1 cm resolution)[44], and new data for MIS 7.5 and MIS 9.3 (8 cm resolution). The concentrations of ssNa$^+$ were measured by classical ion chromatography on discrete samples collected using a melting device connected to an auto-sampler for the MIS 1, MIS 7.5, and MIS 9.3 samples, whereas Continuous Flow Analysis (CFA) was applied for the MIS 5.5 samples[44].

The total deposition ssNa$^+$ flux is calculated by multiplying the measured ice concentration of ssNa$^+$ by the reconstructed

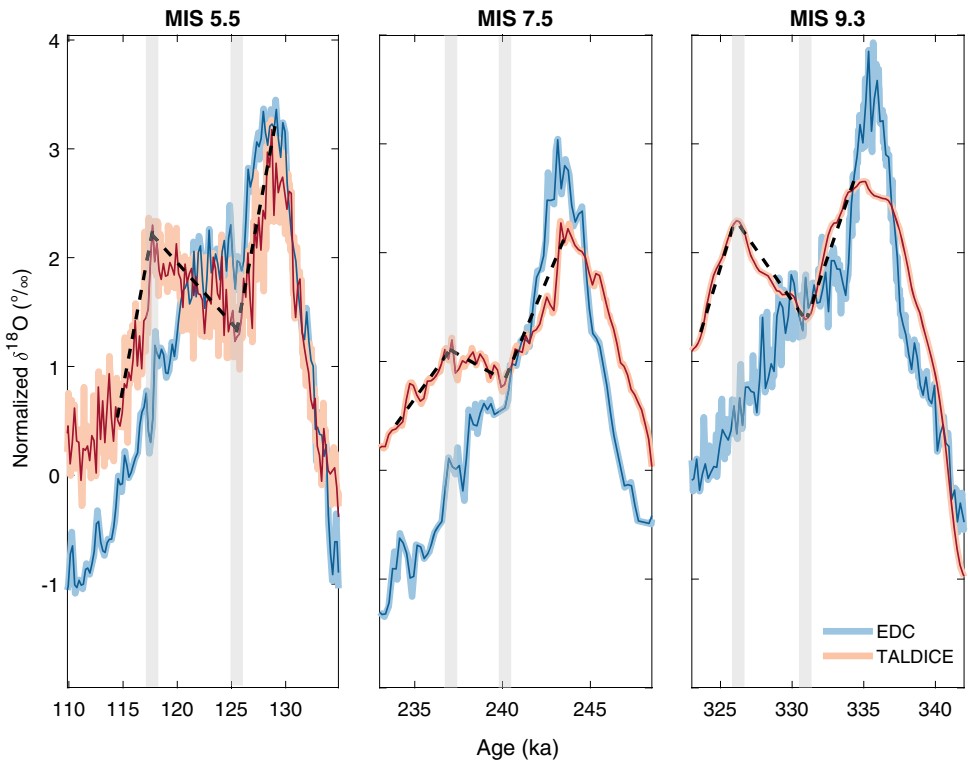

**Fig. 5 | Identification of change points and anomalies in the Talos Dome ice core (TALDICE) in comparison to EPICA Dome C ice core (EDC).** The original TAL-DICE and EDC δ18O records were normalized (pink and light blue, respectively) and then resampled with a 200 year time step (red and dark blue, respectively). Change points in the TALDICE record (grey bars) are identified where the record shows a change in slope (dashed black lines). Anomalies are calculated based on the differences in values in the resampled record between the two change points.

accumulation rates. The accumulation rates are derived from the AICC2012 age scale[66] for the upper part of TALDICE (until 1438 m depth), while for the deepest part (1438–1578 m) the accumulation rates are obtained from the TALDICE-deep1 age scale[23].

### Refined age model for core U1361A

To compare the sedimentological and geochemical records from core U1361A with the TALDICE ice core isotopic records, we need to define a common chronostratigraphy. The original chronology for the U1361 composite splice (U1361A and U1361B) is based on biostratigraphic and magnetostratigraphic data and one radiocarbon date[7,54,67]. The TAL-DICE chronology is based on the AICC2012 age scale until ~150 ka[66], built with a multi-site approach including both Greenland and Ant-arctic ice cores, while for the older part it is based on TALDICE-deep1 chronology[23].

Here we use the AICC2012 ice core chronology[66] as a reference curve in order to compare the late Pleistocene sediment core data from U1361A to the TALDICE ice core record. Specifically, we refine the existing U1361A age model through the alignment of barium/aluminium (Ba/Al) ratios from XRF-scanning[7] with the EDC δD record on the AICC2012 age scale. Glacial-interglacial cycles are clearly expressed in the U1361A Ba/Al record, with higher ratios reflecting warmer conditions and reduced sea-ice extent during interglacials[7]. As such, and similar to the EDC δD record, the Ba/Al record reflects a combination of local forcing and global climate boundary conditions. We apply a conservative tuning strategy to align the two records, using tie points (derived by visual matching) only at the mid-points of the major glacial terminations I-V, where large and rapid signals are recorded in both the U1361A Ba/Al and EDC δD records. Such transitions are likely to be synchronous in the two records, at least at the multi-ka sample resolution of the U1361A marine record, whereas there is greater uncertainty in the exact timing and

progression of productivity changes during the steps from inter-glacial to glacial periods. The U1361A Ba/Al record on its original chronology and on its refined age model is presented in the Sup-plementary Information (Supplementary Fig. 1).

Over the interval of interest for the present study (i.e. ~100–350 ka), the new age model for core U1361 differs by only 0–6 ka compared to the previous age model in which the sedi-mentation rate was assumed to be constant[7]. The temporal resolu-tion of the U1361A record is on the order of several ka for the time period from 0 to 345 ka. Specifically, the mean resolution of the record is 5.9 ka for IBRD, 6.5 ka for Nd isotopes, and 3.4 ka for Ba/Al ratios.

### GRISLI ice sheet model

The GRISLI (Grenoble ice sheet and land ice) model is a large-scale three-dimensional thermomechanical ice sheet model. In this work, we use the GRISLI version 2.0[24] to model changes of the Antarctic ice sheet between 100 and 400 ka, since it is mostly designed for multi-millennial integrations. The model combines an inland ice model with an ice shelf model, extended to the case of ice streams con-sidered as dragging ice shelves. The latest release includes a better representation of grounding line migration and a sub-glacial hydrology model. The model uses finite differences on a Cartesian grid at 5–40 km resolution depending on the application. Here we use a 40 km grid to take advantage of the model calibration per-formed in Quiquet et al.[24]. Given its low numerical cost at this resolution, Quiquet et al.[24] perform an ensemble of 600 simulations to calibrate the mechanical parameters. We use the ensemble member of Tsai et al.[68] for the formulation of the flux at the grounding line, since it best reproduces the glacial-interglacial transitions. This ensemble member is labelled AN40T213[24]. Note that, due to the relatively coarse resolution (40 km) applied here,

fine-scale structures such as individual ice streams might not be properly represented. The model setup used to perform the transient paleo ice sheet simulations is identical to that used by Quiquet et al.[24]. The GRISLI simulations results are presented in the Supplementary Information.

## Data availability

The data generated in this study have been deposited in the PANGAEA database.

The $\delta^{18}O$ data (5 cm resolution for MIS 7.5 and MIS 9.3), d-excess profiles (5 cm resolution for MIS 5.5, MIS 7.5, and MIS 9.3), and ssNa+ flux data (7–8 cm resolution for MIS 5.5, MIS 7.5, and MIS 9.3) from the TALDICE ice core are available at https://doi.pangaea.de/10.1594/PANGAEA.941857. The refined U1361A sediment core age model, the IBRD record (ice rafted debris) and the $^{143}Nd/^{144}Nd$ record are available at https://doi.pangaea.de/10.1594/PANGAEA.941906.The GRISLI 2.0 simulations results generated in this study are available at https://doi.pangaea.de/10.1594/PANGAEA.946775.

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

## Acknowledgements

I.C., B.S. and M.F. disclose support for the research of this work from the TALDEEP project funded by MIUR (PNRA18_00098) and from the joint European program Talos Dome Ice core Project (TALDICE). This is TAL-DICE publication no 61. I.C. also acknowledges funding from the Vinci Scholarship (grant number C2-1017) by the Università Italo-Francese/ Université Franco-Italienne. M.F. also discloses support for the research of this work from the Dept. Science, Roma Tre University (MIUR-Italy Dipartimenti di 588 Eccellenza 2018-2022). A.L. disclose support for the research of this work from the European Research Council under the European Union H2020 Programme (H2020/20192024)/ERC (grant agreement no. 817493 ERC ICORDA). A.Q. acknowledges funding from the SCOR foundation project COASTRISK. D.J.W. discloses support for the research of this work from the NERC independent research fellowship (grant number NE/T011440/1).

## Author contributions

I.C. performed the isotopic analysis, processed the experimental data, designed the main conceptual ideas, wrote the manuscript, and designed the figures. A.Q. performed the model experiments, contributed to the manuscript, and designed the figures. A.L., B.S., and M.F. contributed to the interpretation of results and revised the manuscript. D.J.W. refined the sediment core age model, contributed to the interpretation of results, and revised the manuscript. M.S., F.W., and R.M. performed the chemistry analysis and contributed to the interpretation of results. C.B. revised the manuscript.

## Competing interests

The authors declare no competing interests.
