## [Peer Review File · Nature Communications]

Wilkes Subglacial Basin Ice Sheet response to Southern Ocean Warming During Late Pleistocene InterglacialsReviewers' Comments:

Reviewer #1:

Remarks to the Author:

In the manuscript "Response of the Wilkes Subglacial Basin Ice Sheet to Southern Ocean Warming During Late Pleistocene Interglacials" by Crotti and others, the authors present a new high resolution $\delta^{18}\text{O}$ record of the TALDICE ice core. These data are compared with water isotope measurements from the EDC ice core. The authors focus on interpreting discrepancies between EDC and TALDICE immediately following terminations and conclude that the TALDICE data record ice elevation drops at the Talos dome in response to the warm interglacial conditions, specifically the SO warming. They determine magnitude of elevation changes using 3 different $\delta^{18}\text{O}$ lapse rate estimates. The different lapse rates are used to generate a range of elevation changes which they site as uncertainty of the elevation changes. The plausibility of elevation changes in the Wilkes Basin in response to SO temperatures are evaluated with both an ice sheet model and observations from marine sediments from core collected offshore of Wilkes. I find the paper to be well motivated and the conclusion, if correct, to be extremely significant. This could be a very important contribution. Actual records of what the ice is doing in response to periods of warming are rare. And the authors do exactly what they should do with it--calibrate and ice sheet model and determine the magnitude of loss. This is certainly a topic worthy of appearing in nature communications. I do have some questions about the conclusion that the record is purely from changes in elevation alone. Below I describe a few questions or thoughts that I hope will improve the final product.

When I first read the paper and its central observation-- that one Antarctic ice core has water compositions that differ from another -- I immediately thought about Buizert et al., 2018 (doi: 10.1038/s41586-01-0727-5). And while the focus of the two papers are different I was surprised to see it not referenced. Buizert focuses on the discrepancies in $\delta^{18}\text{O}$ for 5 Antarctic ice core but over a shorter timescale (60-10 ka). Also, while Crotti focuses on the time periods immediately following the 3 last glacial terminations, Buizert focuses on the polar see saw driven changes in SO temperature, so called Antarctic isotopic maximum "AIMs". It might be that because of these differences that Crotti and others did not mention this previous work. But I'll add here that Ai et al., 2020 (DOI: 10.1126/science.abd2115) make the point that SH warming during terminations and AIMs are both product of shifting SH westerlies and increasing SO SST. Because of this, I think in some ways the Buizert and Crotti studies should be comparable but it's difficult for me to say from just studying the figures of the papers alone (I did not download and work with the studies data). In Buizert, the source of the discrepancy in ice core records is probed using principal component analysis. One component, explaining 83% variance in the signal is interpreted as the "oceanic" or classic bipolar seesaw signal that produces a geographically homogenous change in temperature across Antarctica. If this were the only component influencing composition of Antarctic ice core, then I'd tell Crotti and others to feel free interpreting the changes as from elevation. The second component, accounting for 13% of the variance is spatially heterogenous among the different ice core and is interpreted, based on its near coincidence with Greenland ice core records, as an "atmospheric" SH signal. The reason I think that Crotti and coauthors may be interested in this paper, is that this atmospheric forced temperature anomaly is geographically heterogenous effecting Talos dome differently than Dome C. From this observation alone the discrepancies between Talos and EDC must include this atmospheric component. The question is: how much of the discrepancy between Talos and EDC could be from this heterogenous atmospheric component? The variations in $\delta^{18}\text{O}$ resolved by Crotti and others are large, up to 4 ‰ at some time periods, while the temperature anomaly attributed to the atmospheric effect by Buizert is ~ 1 ‰. But this ~ 1 ‰ contribution is for smaller millennial scale SO warming events (e.g. an AIM) not a termination. But Buizert states that the "magnitude of the Antarctic atmospheric response is roughly proportional to the $\delta^{18}\text{O}$ perturbation in Greenland ice cores." If this were to be extended to terminations, than the atmospheric response in Antarctica would be at least 2‰. This extension is based on the idea that terminations are larger, longer AIMs: prolonged warm periods of SO upwelling (Ai et al., 2020, Wolff et al., 2009).

Related specific questions or comments:

- ♣ Could this atmospheric component explain some portion of the observed deviation between Talos dome and Dome C, as reported by Crotti?
- ♣ Would the calculated ice elevation changes reported by Crotti and others be even larger if the data were corrected for this small effect?
- ♣ Here the authors seem to assign significance to the offset between TALDICE and EDC after the largest temperature peaks (line 111-120). But the ice cores differ before peak temperatures as well (Fig. 2). What do the authors think the source of this is? In MIS 5 this is small, the records agree well, but in MIS 7 they are large.
- ♣ On line 109, the authors describe the EDC d18O record as “representative of a common Antarctic signal”. What is meant by this?
- ♣ Observation that the secular changes in deuterium excess are geographically homogenous was also noted in Buizert.
- ♣ The authors discussion of the processes controlling Southern Ocean temperature variations would be improved by incorporating the language developed and used by Ai et al., 2020; Buizert et al., 2018; Wolff et al., 2009. For example, terminations begin because of SO upwelling driven bi-polar seesaw induced shifts in southern westerlies (Ai’s model 3) but warm polar temperatures persist as CO₂ induced warming (Ai’s mode 1) continues to drive shifts in the southern westerlies causing SO upwelling.

Comments on ice sheet model: As a reviewer I’m not able to comment on this specific ice sheet models veracity other than to say they these kinds of models need to be calibrated by the very same kind of geologic data the authors interpret the TALDICE record to reflect. Many modeling papers document the use of SO temperature records inducing Antarctic ice loss. What many of these papers are missing is geologic data to determine how SO temperature couples with the ice. In other words the models are not calibrated. The manuscript here, uses the elevation changes to evaluate model sensitivity tests. This is a positive aspect of the manuscript. However, it’s unclear why the North Atlantic Deep-water record is chosen to drive the ice sheet models. This record is not even presented in figure 3 which is where I was expecting it. Rather an Indian ocean SST record is. I’m not sure why either of these decisions was made. Also why not use an ice core (dome F or EDC) SH temperature record which are mostly explained by SO SST (Andreson et al., 2021; doi.org/10.1016/j.quascirev.2021.106821)? There is a basis for doing so, both Blasco et al 2019 and Golledge et al., 2014 use EDC as a proxy for SO to drive the record. These would be higher resolution than any ocean core record and they are synchronized with Talos dome.

Minor additional comments

- It may benefit the reader if the authors add a sentence to clarifying the sign of the change in d18O for a given elevation? Or edit the figure 2 to aid in understanding heavier isotopes indicates lower elevations. The only place I could find assigning a sign to the generic “elevation changes” is at line 180.
- The resolution of figure 2&3 are too low.

Reviewer #2:

Remarks to the Author:

This paper brings new insight into the retreats of the Antarctic ice sheet in the area of the Wilkes Subglacial Basin during interglacials back to MIS-9.3. The hypothesis testing of the ice core d18O and dD record signal during these interglacials is clearly laid out. I like that the paper combines ice sheet modeling with the ice elevation information from ice core d18O and dD records, this is quite convincing. The integration with the offshore evidence from IODP Site U1361 is also sound, and

supports the main conclusion of the paper, that there was ice retreat of the ice margin in the Wilkes Basin area during MIS 9.3 and MIS 5.5. I have some comments, which include small- and medium-scale changes, after which, in my view, the manuscript will be ready to publish.

Changes:

The paper that describes the dating of the TALDICE ice core, Crotti et al., 2021, ref 50, should be cited much earlier on in the manuscript. There should be text in the introduction to explain how the current manuscript builds on the lead author's 2021 paper, which also includes TALDICE dD results.

Line 82. Replace:

"with late Pleistocene sedimentological and geochemical records from the marine core U1361A offshore Wilkes Subglacial Basin7 (Figure 1)"

with:

"with late Pleistocene sedimentological and geochemical records from International Ocean Discovery Program (IODP) Hole U1361A, offshore of the Wilkes Subglacial Basin7 (Figure 1)"

Figure 1. IODP Site U1361 is misplaced. Currently marked on the continental shelf, it should be further north on the continental rise.

Line 342 – Figure 3f, not 4f.

Line 349 – "ice sheet collapse" during MIS 11 – better to specify what is meant here – i.e. not the whole ice sheet collapsing but probably an ice margin retreat further into the Wilkes Basin than during subsequent interglacials. (In fact I'd speculate that the absence of clear MIS-11 ice at TALDICE might be seen as evidence of ice retreat a good ways back into the basin.)

Add a figure to the supplement to illustrate the updated time scale for U1361. Perhaps including the original time scale and the new correlation to EDC AICC 2012.

Figure 3 shows the position of the retreated grounding line in the ice sheet model results, and Fig 8 the overall ice volume change. I'd like to see a more detailed map of where the ice is drawn down, e.g. with contour lines. This would be informative to include. Probably as an extra figure in the supporting information.

Notes to mention in the discussion, and perhaps follow up with in follow-up work:

Neither of the two sets of models in the manuscript, which start in a glacial or interglacial state at 400 ka, MIS-11, represent the likely state of the ice sheet at that time, when the ice sheet was probably more retreated than at present. I would be interested to see the equivalent models for this starting condition. Still, the authors present the existing models clearly, and such further modeling can be future work.

The double peaks in d18O during MIS 9.3 and 5.5 are described in the results and compared to recent sea surface temperature results from the South Indian Ocean (Fig 3f). I think the authors can go further with their observations here. In both cases the second peak is about ~10 kyr after the first, and close to the time of peak southern hemisphere insolation. This hints that local insolation is a factor in Antarctic ice margin stability during these interglacials. It has been hypothesized for a while now that Antarctic ice responded to local insolation, as a way to explain the 40 kyr cyclicity from ~3 to 1 Ma (e.g. Raymo, Lisiecki, and Nisancioglu, 2006). Anyway, it is very interesting to find suggestions of local insolation control in the last 1 Myr, it might serve as a hypothesis for a future paper.

AUTHORS COMMENTS

The authors thank the reviewers for their constructive and insightful comments and corrections that will increase the scientific quality of the manuscript and its clarity.

Here we present our answers to the reviewers and comments. In particular, we produce an extended analysis of the atmospheric contribution to the TALDICE isotopic signal during past interglacial periods, including performing Principal Component Analysis. In addition, we provide new GRISLI simulations, changing the oceanic forcing and the Antarctic ice sheet initial conditions, according to the reviewers' suggestions. One of the main differences between the submitted manuscript and the revised one is the introduction of the new Deglaciated State (DS) simulations, which are more representative of the Antarctic Ice sheet state at 400 ka, and the production of the Supplementary Information file.

We also perform corrections on the manuscript according to the reviewers' advice, such as introducing additional references and implementing their suggestions in the discussion section.

We hope that the revised version of the manuscript improves the quality of the text and of the scientific message. Please find below our answers to the reviewers' comments.

Reviewer #1 (Remarks to the Author):

Changes in response to **Reviewer #1** comments are highlighted in **light blue** in the tracked-record version of the manuscript.

In the manuscript "Response of the Wilkes Subglacial Basin Ice Sheet to Southern Ocean Warming During Late Pleistocene Interglacials" by Crotti and others, the authors present a new high resolution d18O record of the TALDICE ice core. These data are compared with water isotope measurements from the EDC ice core. The authors focus on interpreting discrepancies between EDC and TALDICE immediately following terminations and conclude that the TALDICE data record ice elevation drops at the Talos dome in response to the warm interglacial conditions, specifically the SO warming. They determine magnitude of elevation changes using 3 different d18O lapse rate estimates. The different lapse rates are used to generate a range of elevation changes which they site as uncertainty of the elevation changes. The plausibility of elevation changes in the Wilkes Basin in response to SO temperatures are evaluated with both an ice sheet model and observations from marine sediments from core collected offshore of Wilkes. I find the paper to be well motivated and the conclusion, if correct, to be extremely significant. This could be a very important contribution. Actual records of what the ice is doing in response to periods of warming are rare. And the authors do exactly what they should do with it-calibrate and ice sheet model and determine the magnitude of loss. This is certainly a topic worthy of appearing in nature communications. I do have some questions about the conclusion that the record is purely from changes in elevation alone. Below I describe a few questions or thoughts that I hope will improve the final product.

When I first read the paper and its central observation-- that one Antarctic ice core has water compositions that differ from another -- I immediately thought about Buizert et al., 2018 (doi: 10.1038/s41586-01-0727-5). And while the focus of the two papers are different I was

surprised to see it not referenced. Buizert focuses on the discrepancies in $\delta^{18}O$ for 5 Antarctic ice core but over a shorter timescale (60-10 ka). Also, while Crotti focuses on the time periods immediately following the 3 last glacial terminations, Buizert focuses on the polar see saw driven changes in SO temperature, so called Antarctic isotopic maximum “AIMs”. It might be that because of these differences that Crotti and others did not mention this previous work.

But I’ll add here that Ai et al., 2020 (DOI: 10.1126/science.abd2115) make the point that SH warming during terminations and AIMs are both product of shifting SH westerlies and increasing SO SST.

Because of this, I think in some ways the Buizert and Crotti studies should be comparable but it’s difficult for me to say from just studying the figures of the papers alone (I did not download and work with the studies data). In Buizert, the source of the discrepancy in ice core records is probed using principal component analysis. One component, explaining 83% variance in the signal is interpreted as the “oceanic” or classic bipolar seesaw signal that produces a geographically homogenous change in temperature across Antarctica. If this were the only component influencing composition of Antarctic ice core, then I’d tell Crotti and others to feel free interpreting the changes as from elevation. The second component, accounting for 13% of the variance is spatially heterogenous among the different ice core and is interpreted, based on its near coincidence with Greenland ice core records, as an “atmospheric” SH signal. The reason I think that Crotti and coauthors may be interested in this paper, is that this atmospheric forced temperature anomaly is geographically heterogenous effecting Talos dome differently than Dome C. From this observation alone the discrepancies between Talos and EDC must include this atmospheric component.

The question is: how much of the discrepancy between Talos and EDC could be from this heterogenous atmospheric component?

ANSWER 1

The comment of the reviewer addresses an interesting point that we have also considered while writing the manuscript and interpreting the data. The main reasons for not including the reference in the first version of the manuscript were that Buizert et al. focuses on different time periods (60-10 ka versus 115-350 ka) and the warming events were of different magnitude (Antarctica Isotopic Maxima are minor warming events in comparison to interglacial stages). However, considering the suggestion of the reviewer, we believe that adding this reference helps provide a more complete and robust interpretation of the TALDICE isotopic signal. We added the reference at lines 144-145 of the manuscript: “consistent patterns during interglacial stages (Figure 2b), as already observed for shorter warming events (Buizert et al. 2018)”. Still, it should be considered that while we observe differences in the interglacial signals between TALDICE and EDC, Buizert et al. find very similar patterns between the two sites over AIM events, so the comparison between the two studies is not straightforward.

The Buizert et al. 2018 study shares some questions with our manuscript, since it aimed to investigate the isotopic response to warming events (AIM events in this case) in Antarctic ice cores between 10 ka and 60 ka. Applying a Principal Component Analysis (PCA) approach to the isotopic signal of 5 Antarctic ice cores (TALDICE, EDC, Dome Fuji, EPICA Dronning Maud Land, West Antarctic Ice Sheet Divide), Buizert et al. identify that the majority of the

Antarctic isotopic signal variance during the AIM events is explained by the oceanic component (83%), while 13% of the variance is explained by the atmospheric signal on a 1500 a window. In Figure 2g of the Buizert paper, the atmospheric component (PC2) is homogeneous in East Antarctica (EDC and TALDICE), while it is heterogeneous for the whole Antarctic continent, with the largest difference over AIM events being observed between EDML and the other sites. Buizert et al. state that during AIMs, TALDICE and EDC isotopic signals are influenced by similar oceanic and atmospheric patterns, indicating homogeneous variations in East Antarctica. To conclude, in Buizert et al. (2018), the atmospheric and oceanic components (PC1 and PC2) are spatially heterogeneous among different Antarctic cores, but homogeneous for East Antarctica (Figure 2). According to the results of Buizert et al. (2018), a similar behaviour of the TALDICE and EDC isotopic signals is expected during warm periods considering the oceanic and the atmospheric components, which differs from our main observation of spatial variability during interglacial periods.

In our manuscript, we investigate the signal of the source evaporative region (transmitted by atmospheric transportation) by comparing the d-excess records of TALDICE and EDC during past interglacials without finding any clear difference. According to the suggestions of the reviewer and to test the presence and the importance of the atmospheric component in the TALDICE signal, we perform a PCA on the $\delta^{18}\text{O}$ records of EDC (Landais et al., 2021), Dome Fuji (Uemura et al. 2018), Vostok (Vimeux et al. 2001) and TALDICE during MIS 5.5, 7.5 and 9.3 on AICC2012 age scale (Bazin et al. 2013). We follow a similar PCA approach to the one applied by Masson Delmotte et al. (2011). We place the TALDICE, EDC and Dome Fuji $\delta^{18}\text{O}$ records on the AICC2012 common age scale (Bazin et al. 2013) and re-sample on a 300 a time step. These re-sampled data are then smoothed using a 5-point binomial filter. The PCA is performed with the MATLAB *pca* function and the results are reported in Table 1R.

Period	% of variance	Coefficients (or loadings)			
		TALDICE	Dome F	EDC	Vostok
MIS 5.5	PC1= 95.6503	0.2649	0.5952	0.4898	0.5793
	PC2= 2.7950	0.7411	-0.1434	0.3938	-0.5245
MIS 7.5	PC1= 99.1046	0.2231	0.5781	0.5447	0.5650
	PC2=0.7418	0.8280	-0.5275	0.1874	0.0322
MIS 9.3	PC1= 90.6571	0.3180	0.5702	0.5297	0.5415
	PC2= 6.9751	0.8008	-0.2071	0.2545	-0.5011

Table 1R: Results of PCA performed on TALDICE, Dome F, EDC and Vostok $\delta^{18}\text{O}$ records during MIS 5.5, 7.5 and 9.3. The percentages of explained variance for PC1 and PC2 are reported with the respective PCA coefficients (or loadings).

The PCA results indicate that more than 90% of the signal is explained by the first component PC1, while the second component PC2 accounts for only ~7% or less. Our results are comparable to the results obtained by Masson Delmotte et al. (2011) for MIS 5.5, where up to 90% of the isotopic signal is captured by the PC1. The first component is interpreted as a common $\delta^{18}\text{O}$ Antarctic signal, while the second component is related to site-specific differences. The variance explained by PC1 for interglacials is larger in comparison to the value obtained by Buizert et al. during AIMs (83%). The second component accounts for ~7% or less during interglacials, indicating that inter-site differences explain a small portion of the isotopic signal. By comparison, 13% of the variance is explained by PC2 during AIMs in Buizert et al. (2018).

In addition, we calculate the coefficients (or loadings) for the different sites in Table 1R for PC1 and PC2 for all the cores. Positive loadings indicate that a variable and a principal component are positively correlated: an increase in one results in an increase in the other. Negative loadings indicate a negative correlation. Large (either positive or negative) loadings indicate that a variable has a strong effect on that principal component. Our PCA highlights that TALDICE is clearly an outlier in comparison to the other cores (Vostok, EDC and Dome F) which show homogeneous values (Table 1R). The loadings associated with PC1 for TALDICE are characterized by values of 0.2-0.3 for the three interglacials, while all the other sites show spatially homogeneous values of ~ 0.5 during all interglacials. To better interpret the signal explained by the PC1 we compute the scores for PC1 and PC2 and display their values on the AICC2012 age scale (Figure 1R). In Figure 1R the normalized $\delta^{18}\text{O}$ profiles of TALDICE, Vostok, EDC and Dome F (data centred and scaled to have mean 0 and standard deviation 1) during interglacials MIS 5.5, 7.5 and 9.3 and the scores of PC1 and PC2 are compared. The score plot shows that PC1 clearly represents the early interglacial isotopic maxima signal observed in all cores, but not the second isotopic maximum observed in the TALDICE $\delta^{18}\text{O}$ record. PC1 has a very similar evolution to the $\delta^{18}\text{O}$ records of Vostok, EDC and Dome F. As for PC2 capturing the second interglacial peak, which is only seen at TALDICE, the associated loading values show heterogeneous values among all the sites, especially between EDC and TALDICE.

The results of our PCA analysis are very different from the one performed on AIM events by Buizert et al. (2018). First, the heterogeneous loading obtained for PC1 during interglacials in our study (for TALDICE compared to the other East Antarctic cores) does not agree with the findings of Buizert et al., where PC1 has homogeneous coefficients for all the Antarctic cores (Figure 2). Buizert et al. (2018) also find similar PC2 values for TALDICE and EDC during AIM events which is opposite to our results over interglacial periods. Because of such contrasting results, we prefer not to perform a deep comparison with Buizert et al. (2018) and consider that, in our study, PC2 does not necessarily represent the atmospheric component. The lack of homogeneous loadings between TALDICE and EDC and the similarity of the shape between PC2 and the TALDICE $\delta^{18}\text{O}$ signal support that here the PC2 is not connected to the atmospheric mode presented in Buizert et al. 2018.

Figure 1R: Top, normalized $\delta^{18}\text{O}$ values (data centred and scaled to have mean 0 and standard deviation 1) for TALDICE (blue), Vostok (red), EDC (yellow), and Dome F (purple) on AICC2012 age scale to better compare the isotopic profiles. Bottom, scores of PC1 (blue) and PC2 (red) plotted on AICC2012 are helpful to evaluate the behaviour of the components and interpret the isotopic data.

The variations in $\delta^{18}\text{O}$ resolved by Crotti and others are large, up to 4 ‰ at some time periods, while the temperature anomaly attributed to the atmospheric effect by Buizert is $\sim 1\%$. But this $\sim 1\%$ contribution is for smaller millennial scale SO warming events (e.g. an AIM) not a termination. But Buizert states that the “magnitude of the Antarctic atmospheric response is roughly proportional to the $\delta^{18}\text{O}$ perturbation in Greenland ice cores”. If this were to be extended to terminations, then the atmospheric response in Antarctica would be at least 2%. This extension is based on the idea that terminations are larger, longer AIMS: prolonged warm periods of SO upwelling (Ai et al., 2020, Wolff et al., 2009).

ANSWER 2

Buizert et al. (2018) indeed investigate the AIMS associated with $\delta^{18}\text{O}$ increases of $\sim 1\%$ between 10 ka and 60 ka. In this study, the temperature anomaly attributed to the atmospheric effect is smaller and of the order of $+0.2 - +0.3\%$ in TALDICE and EDC (Figure 2h of the Buizert et al. paper).

The early interglacial isotopic maxima variations at TALDICE during MIS 5.5, 7.5 and 9.3 are of the order of up to $+4\%$ as indicated by the reviewer and we agree that they can be linked to prolonged AIM. However, our study is not focused on the early interglacial isotopic maxima. We are rather investigating the TALDICE $\delta^{18}\text{O}$ anomalies during the late interglacials prior to glacial inception. These anomalies (Table 1 of the manuscript and section “Identification of change-points and calculation of anomalies” of the Methods) are of the order of $+0.68\%$ to $+1.68\%$ (Table 2R), i.e. much larger than the atmospheric effect noted by Buizert et al. (2018) and not associated with the strong and early interglacial maximum.

As a consequence, while we agree with the reviewer that we should quote the study of Buizert et al. (2018) in the new version of the manuscript (line 145), we consider that the two studies focus on different regional climate and ice sheet systems, since TALDICE is more influenced by local conditions in comparison to the other Plateau sites.

	MIS 5.5	MIS 7.5	MIS 9.3
Time interval of $\delta^{18}\text{O}$ anomaly (ka) at TALDICE	117-127	237-240	326-331
$\delta^{18}\text{O}$ max and $\delta^{18}\text{O}$ min (‰)	(-35.45)-(-37.13)	(-37.19)-(-37.87)	(-35.46)-(-36.89)
$\Delta\delta^{18}\text{O}$ (‰)	+1.68	+0.68	+1.42

Table 2R: Isotopic anomalies calculated for Talos Dome during interglacial MIS 5.5, 7.5, and 9.3. Isotopic anomalies are calculated from the TALDICE $\delta^{18}\text{O}$ record resampled at 200 year intervals.

Related specific questions or comments: Could this atmospheric component explain some portion of the observed deviation between Talos dome and Dome C, as reported by Crotti?

ANSWER 3

Following the reviewer observations, in ANSWER 1 we tried to improve our analysis and investigate the $\delta^{18}\text{O}$ signal during interglacials for TALDICE, EDC, Vostok and Dome Fuji. We performed PCA on the isotopic records in order to compare PC1 and PC2 loadings and scores for TALDICE, EDC, Dome Fuji and Vostok cores during the past interglacials, following the same approach presented in Buizert et al. (2018). Our results show that the PC1 shows a triangular shape of the isotope maximum events as stated in Buizert et al. (2018) for all the interglacials, with spatially homogeneous values, with the exception of TALDICE. The second component (PC2) shows a double peak function for MIS 5.5 and MIS 9.3, different from the step-like function observed by Buizert et al. for AIMs, with a heterogeneous spatial pattern during the past interglacials.

According to our results it is very unlikely that PC2 calculated for interglacials is representative of the atmospheric component for several reasons:

- i. It shows a different signal (double peak reflecting the isotopic anomaly in TALDICE) in comparison to the PC2 computed by Buizert, which reflects atmospheric circulation changes connected to the Southern Annular Mode (SAM) and Pacific-South American (PSA) patterns.
- ii. The variance attributed to the second component is very limited (maximum ~7%) for all the interglacials, while in the Buizert et al. work it is 13%.
- iii. PC2 for TALDICE shows similar values during the interglacials, but TALDICE is an outlier in comparison to the other cores. This is very different from the results obtained by Buizert et al., where PC2 is characterized by a spatially homogeneous pattern for East Antarctica.

To conclude, the PC2 in TALDICE cannot be explained by an East-Antarctic regional atmospheric component because it mainly reflects the observed deviation in the isotopic record between TALDICE and EDC during the second half of the past interglacials. We conclude that, while the atmospheric component plays a role during the AIMs events, it is not responsible for the $\delta^{18}\text{O}$ anomaly observed at TALDICE during the past interglacials.

Would the calculated ice elevation changes reported by Crotti and others be even larger if the data were corrected for this small effect?

ANSWER 4

We agree with the reviewer that the “real” elevation changes at Talos Dome were probably larger than the ones calculated from the $\delta^{18}\text{O}$ TALDICE anomalies during late interglacials. We suggest that the elevation changes at Talos Dome were probably larger in comparison to the calculated variations especially for MIS 9.3. However, we do not consider the atmospheric effect to drive this underestimation (since it is negligible according to our results), but the thinning and diffusion processes acting on the isotopic signal in the deeper part of the core. This is stated in lines 281-285 of the manuscript: “However, the ice thickness variations simulated at Talos Dome during MIS 9.3 are much greater than those calculated from the isotopic record (~100-250 m) (Table 1). The TALDICE lower temporal resolution (~200 years/5 cm) below 1530 depth probably leads to a smoothing of the isotopic signal, which could be responsible for a muted $\delta^{18}\text{O}$ signal during MIS 9.3”.

In Figure 5 of the manuscript, it is seen that the TALDICE $\delta^{18}\text{O}$ raw record during MIS 9.3 (pink line) shows an extremely reduced variability in comparison to the raw data of MIS 5.5. In fact, the raw data resolution decreases from 32 a / 5 cm for MIS 5.5 to 192 a / 5cm for MIS 9.3. Such reduction of the temporal resolution, due to the thinning for the oldest portion of TALDICE, have smoothed the interglacial isotopic maxima signal and the anomaly in the late part of the MIS 9.3.

The GRISLI GS-5 simulation (the one that better agrees with elevation changes calculated at Talos Dome from the isotopic record) models an elevation decrease at Talos Dome during MIS 9.3 of ~-470 m, larger in comparison to the computed ~-100 - -250 m from the isotopic anomaly of + 1.42 ‰. To obtain the modelled elevation, the isotopic anomaly in the late stage of MIS 9.3 at TALDICE should fall in the range of +2.5 - + 6.3 ‰, taking into account the different lapse rates in Table 1.

Here the authors seem to assign significance to the offset between TALDICE and EDC after the largest temperature peaks (line 111-120). But the ice cores differ before peak temperatures as well (Fig. 2). What do the authors think the source of this is? In MIS 5 this is small, the records agree well, but in MIS 7 they are large.

ANSWER 5

We thank the reviewer for raising this interesting question and we agree that this point should be clarified in the manuscript. The difference before the peak temperatures observed in MIS 7.5 between TALDICE and EDC is related to the definition of the TALDICE deep1 age scale (Crotti et al. 2021). In particular, the age scale is based on ice and gas markers obtained from the TALDICE and EDC synchronization of δD and $\delta^{18}\text{O}_{\text{atm}}$ records (Figure 2R) as input of the IceChrono1 model (Parrenin et al. 2015). The final chronology is affected by the $\delta^{18}\text{O}_{\text{atm}}$ synchronization uncertainties due to the low resolution of the TALDICE $\delta^{18}\text{O}_{\text{atm}}$ record (1.55 m, ~2620 years).

In particular, for the depth range between 1516 m to 1529 m, corresponding to MIS 7.5 and MIS 8, the low resolution of the $\delta^{18}\text{O}_{\text{atm}}$ TALDICE record does not allow a perfect match with the EDC record. This mismatch produces the offset observed by the reviewer.

To obtain a more accurate synchronization and avoid lags between TALDICE and EDC isotopic records, such as the one observed for MIS 7.5, more $\delta^{18}\text{O}_{\text{atm}}$ measurements would be needed in order to reduce the uncertainty associated with the TALDICE deep1 chronology. We add the following sentence at lines 128-133: “In the case of MIS 7.5, the EDC and TALDICE isotopic record diverge also before the interglacial maxima, around 245 ka. However, this offset is probably due to the uncertainties associated with the TALDICE deep1 chronology driven by the low resolution of the $\delta^{18}\text{O}_{\text{atm}}$ record and should not be interpreted as a physical signal (Crotti et al. 2021). A new set of high-resolution $\delta^{18}\text{O}_{\text{atm}}$ measurements would be needed to further investigate this point”.

Figure 2R: Synchronization of TALDICE and EDC $\delta^{18}\text{O}_{\text{atm}}$ records between 150 and 336 ka. (a) TALDICE (blue curves) $\delta^{18}\text{O}_{\text{atm}}$ record and gas tie points (red dots) synchronized on EDC (black curve) (Extier et al., 2018 and new measurements) drawn on the AICC2012 gas age timescale (Bazin et al., 2013). From Crotti et al. (2021).

On line 109, the authors describe the EDC d18O record as “representative of a common Antarctic signal”. What is meant by this?

ANSWER 6

We agree with the reviewer that this sentence is not clear. We want to highlight the concept that the EDC ice core is representative of Antarctic Plateau conditions, and less influenced by local variations in surface elevation or regional sea ice extent. We replaced the sentence “representative of a common Antarctic signal” with the sentence “representative of the Antarctic plateau conditions” at lines 117-118.

Observation that the secular changes in deuterium excess are geographically homogenous was also noted in Buizert.

ANSWER 7

We agree with the reviewer that during warm periods (AIMs and interglacials), the *d*-excess signal for TALDICE and EDC are spatially homogeneous. Following the reviewer’s advice

we add the reference to the Buizert et al. work in lines 144-145 of the manuscript: “as already observed for shorter warming events (Buizert et al. 2018)”.

Moreover, to better specify the characteristics of the d -excess signal we add the following sentence at lines 144-148 with the aim of providing a more complete explanation: “consistent patterns during all interglacials (Figure 2b), as already observed for shorter warming events (Buizert et al. 2018). In particular, the increases in d -excess for TALDICE and EDC during MIS 5.5, MIS 7.5, and MIS 9.3 occur over longer time intervals than the $\delta^{18}\text{O}$ increase, and the maxima in d -excess are reached in the second half of the interglacial period (Landais, Stenni, Masson Demotte, et al. 2021) but before the second $\delta^{18}\text{O}$ anomaly in the TALDICE record”.

In addition, Buizert et al. (2018) state that the similarity between d_{ln} and PC2 of $\delta^{18}\text{O}$ observed in Figure 4a (of the Buizert et al. paper) represents “distinct but consistent manifestations of atmospheric circulation change”. To perform a complete comparison with the Buizert work, we replicate the same experiment with TALDICE data, calculating the d_{ln} for the past interglacials (Figure 3R). The results (Figure 3R) show that we cannot match the double peak shape of PC2 with the d_{ln} signal for MIS 5.5 and 9.3 because the two profiles are very different. Such absence of similarities between PC2 and d_{ln} time evolution further indicates that the anomalies in the TALDICE $\delta^{18}\text{O}$ signal during interglacials are driven by a different influence than the atmospheric signal invoked by Buizert et al. (2018).

Figure 3R: d_{ln} profiles of TALDICE for raw (pink) and resampled (red) d -excess data during past interglacials.

The authors discussion of the processes controlling Southern Ocean temperature variations would be improved by incorporating the language developed and used by Ai et al., 2020; Buizert et al., 2018; Wolff et al., 2009. For example, terminations begin because of SO upwelling driven bi-polar seesaw induced shifts in southern westerlies (Ai’s model 3) but warm polar temperatures persist as CO_2 induced warming (Ai’s mode 1) continues to drive shifts in the southern westerlies causing SO upwelling.

ANSWER 8

We thank the reviewer for their suggestion of improving the discussion with the introduction of specific terminology. We have adopted the proposed language and references accordingly. In particular, we add the following statements in the discussion section at lines 370-378:

“During interglacial periods the combination of the increasing insolation and the decreasing ice sheet volume produce the steepening of the meridional temperature gradient, which strengthens and shifts poleward the Southern Westerly Winds, promoting the upwelling in the Southern Ocean (Ai et al. 2020; Buizert et al. 2018). In addition, deglacial weakening of the Atlantic Meridional Overturning Circulation may have led to warmer Southern Ocean temperatures through the bipolar seesaw mechanism, with early interglacial atmospheric warming also attributable to CO₂ escape from deep waters (Landais, Stenni, Masson Demotte, et al. 2021; Shukla, Crosta, and Ikehara 2021; Wolff, Fischer, and Röthlisberger 2009). The more southerly and warmer Antarctic Circumpolar Current could have induced a dynamic response in the EAIS during MIS 5.5 (Chadwick et al. 2020; Fogwill et al. 2014) and a comparable scenario can also be anticipated for MIS 9.3.”

Comments on ice sheet model: As a reviewer I'm not able to comment on this specific ice sheet models veracity other than to say they these kinds of models need to be calibrated by the very same kind of geologic data the authors interpret the TALDICE record to reflect. Many modelling papers document the use of SO temperature records inducing Antarctic ice loss. What many of these papers are missing is geologic data to determine how SO temperature couples with the ice. In other words the models are not calibrated. The manuscript here, uses the elevation changes to evaluate model sensitivity tests. This is a positive aspect of the manuscript.

However, it's unclear why the North Atlantic Deep-water record is chosen to drive the ice sheet models. This record is not even presented in figure which is where I was expecting it. Rather an Indian ocean SST record is. I'm not sure why either of these decisions was made.

ANSWER 9

The reviewer is touching on an interesting point that was discussed among the authors before taking the decision to use the North Atlantic Deep Water temperature record to represent oceanic forcing.

The Antarctic grounding line migration is primarily driven by sub-shelf melt rates, more specifically in the vicinity of the grounding line. Sub-shelf melt rates are spatially very heterogeneous as they depend on water mass circulation around Antarctica and local topographic features. This process is, at present, the main model uncertainty for the representation of present-day Antarctica and its future evolution (Asay-Davis, Jourdain, and Nakayama 2017; Seroussi et al. 2020). Given this complexity, here we chose a simple approach in which the present-day melt rates are homogeneously perturbed by a palaeoclimatic oceanic index. In doing so, we disregard any geographic changes in water mass distribution around Antarctica, since the modelled changes would be synchronous and of the same relative amplitude for all ice shelves. As the reviewer rightly points out, the choice of the oceanic index is critical in this formulation and that is in partly why we performed sensitivity experiments in which we increased the sub-shelf melt rate.

Ideally, this index should be representative of sub-surface temperature variations, since Antarctic grounding lines are typically located about 500 m below sea level. This prevents the use of a sea surface temperature record since it would be only weakly correlated with sub-surface temperature (Quiquet et al. 2018) and more correlated with atmospheric temperature. Another constraint for this oceanic index is that it has to be continuous for the whole last 400 ka and provide a reasonable temporal resolution. To our knowledge, there is no available record that meets these criteria (i.e. representativity, temporal coverage, and resolution). However, if we have overlooked such a record, we would be happy to include it in our ice sheet modelling as it would be a great addition for this work and future studies. Instead, here we chose to use the deep-sea temperature record derived from benthic foraminifera in the North Atlantic site ODP 980, which represents the North Atlantic Deep Water (NADW) temperature (Waelbroeck et al. 2002). The NADW is first produced by deep convection of dense waters at high northern latitudes, before flowing southward in the deep Atlantic Ocean and upwelling in the Southern Ocean. The upwelled waters can either migrate equatorwards or polewards to sink again to form Antarctic Bottom Water (Ferrari et al. 2014). NADW temperature is thus a reasonable indicator for the available oceanic heat in Antarctic coastal regions.

Following the contribution of the reviewer, we improve the explanation regarding the justification of our choice in the revised version of the manuscript between lines 223 and 237: “The model is forced, for the past 400 ka, by near-surface air temperatures over Antarctica deduced from the EDC δD record (Jouzel et al. 2007; Quiquet et al. 2018), which represents a boundary condition at the surface of the ice sheet for thermomechanical coupling and drives local changes in precipitation and surface mass balance. On the other hand, the oceanic forcing is produced through modification of the present-day sub-shelf melt rate by a palaeo-oceanic index based on the ODP 980 benthic temperatures in the North Atlantic Ocean (Quiquet et al. 2018; Waelbroeck et al. 2002), which represent North Atlantic Deep Water (NADW) temperatures. The NADW is produced by deep convection of dense waters at high northern latitudes, then flows southward in the deep Atlantic Ocean, and upwells in the SO. Due to the lack of a suitable SO proxy record for sub-surface oceanic temperatures spanning the last 400 ka, this estimate of upwelled NADW temperature is a reasonable indicator for the oceanic heat available in Antarctic coastal regions that affects the sub-shelf melting rate at the grounding line. We have also performed additional sensitivity tests using two other estimates of the sub-surface oceanic conditions in the SO (Supplementary Figure 2) and those results are presented in the Supplementary Information”.

Following the reviewer’s advice, we have modified Figure 3 by substituting the DRC 1PC Diatom SST ($^{\circ}C$) (Indian Ocean record) with the ODP 980 record. We had decided to include the DRC record since it is cited in the discussion, but we agree that it is more informative for the reader to see the ODP 980 record employed as ocean forcing.

Also why not use an ice core (dome F or EDC) SH temperature record which are mostly explained by SO SST (Andreson et al., 2021; doi.org/10.1016/j.quascirev.2021.106821)? There is a basis for doing so, both Blasco et al 2019 and Golledge et al., 2014 use EDC as a proxy for SO to drive the record. These would be higher resolution than any ocean core record and they are synchronized with Talos dome.

ANSWER 10

We completely agree with the reviewer that ideally it would be better to force the GRISLI model with a local oceanic record, but unfortunately there is no record currently available of sub-shelf temperatures in the SO spanning the past 400 ka, as introduced in ANSWER 9.

Ice shelves and resulting grounding line dynamics are not responding to SST since there is an important decoupling between surface and sub-surface conditions. Although some authors have used SST (Golledge et al. 2014), or even an atmospheric record such as EDC (Blasco et al. 2019), as a proxy for the ocean forcing in the sub-shelf melt model, we do not think that there is a robust physical justification for doing so. The only advantage of using SST or EDC records is that they cover a wide temporal time frame with good age resolution.

Figure 4R: Oceanic forcing indexes for the GRISLI simulations over the past 400 ka. The three indexes were derived from the ODP 980 bottom water temperatures (Waelbroeck et al. 2002) as applied in the original model simulation (Quiquet et al. 2018) (blue curve), the EDC δD profile (Jouzel et al. 2007) (dark red curve), and the LR04 benthic $\delta 18O$ stack (Lisiecki and Raymo 2005) (black curve).

Nonetheless, to produce a more comprehensive set of simulations as suggested by the reviewer, we perform 18 new experiments using two new paleo-oceanic indexes to force our ice sheet model, starting from GS, IS, and DS initial Antarctic ice sheet state. In a first set of experiments we use an index derived from the EPICA Dome C δD record (Jouzel et al. 2007). In a second set of experiments, we derive our index from the LR04 benthic oxygen isotope stack of Lisiecki and Raymo (2005), which represents a combination of deep-ocean temperature and global (mainly Northern Hemisphere) ice volume. The oceanic forcing indexes for GRISLI simulations are shown in Figure 4R. For both the EDC and LR04 indexes, we use a conversion factor so that the amplitude of the sub-shelf melt change from the last glacial maximum to the present-day is similar to our original index, since the model was calibrated in Quiquet et al. (2018) using this index. As for our reference simulations, for

the two new paleo-oceanic indexes, we perform again three melt scenarios (standard, +5%, and +10%) for three initial ice sheet conditions (glacial, interglacial and deglaciated states), resulting in 18 new experiments. The deglaciated state is introduced according to ANSWER 18 to Reviewer #2.

In order to assess the differences between the sensitivity tests forced by the NADW record (Quiquet et al. 2018), the LR04 record, and the EDC record (Jouzel et al. 2007), we compare (i) the elevation changes at Talos Dome simulated with GRISLI over the past 350 ka (Figure 5R) and (ii) the Wilkes Subglacial Basin ice volume changes over the past 350 ka (figure 6R). Elevation variations at Talos Dome during interglacials are calculated for all the simulations and shown in Table 3R. We can draw two conclusions from these additional experiments:

- First: with the conversion factor used here, the new indexes derived from the EDC and LR04 records produce larger retreat during MIS 5.5 and MIS 9.3 in comparison to the simulations forced by the ODP 980 record. This finding can be explained because the new indexes show (i) strongest peak interglacials with respect to ODP 980 (EDC index) or (ii) higher values during glacial periods (LR04 index) (Figure 4R). However, to circumvent this issue we could have used an alternative conversion factor to limit the sub-shelf melt during interglacials using these new forcings.

- Second: the general pattern of retreat is consistent amongst the different forcing scenarios. There is a larger retreat during MIS 9.3 than during MIS 5.5. However, most of the new experiments produce a strong ice thinning at Talos Dome and large ice volume decrease in the Wilkes Subglacial Basin that is irreconcilable with our data. For these reasons, we present in the manuscript the model runs forced by the NADW record as proposed in Quiquet et al. (2018). Nevertheless, the simulation results obtained applying the EDC and LR04 records as oceanic forcing represent valuable results to include in our publication, so the GRISLI simulations results (ice thickness variation at Talos Dome and WSB ice volume) are reported in the Supplementary Information. In particular, we show the GRISLI simulation results starting from Glacial State (GS), Interglacial State (IS) and from the new Deglaciated State (DS) introduced in the manuscript (see Answer 18 to Reviewer #2) forced by the EDC and LR04 oceanic indexes. The following statement is added at line 234-237 regarding the simulations presented in the Supplementary Information: “We have also performed additional sensitivity tests using two other estimates of the subsurface oceanic water in the SO and the results are gathered in the Supplementary Information”.

Ocean forcing		GRISLI elevation changes at Talos Dome (m)		
Time interval for max elevation anomaly at TALDICE (ka)		115-128	233-241	321-332
IS	Quiquet et al. (2018)	-132	-103	-749
IS-5	Quiquet et al. (2018) +5%	-126	-101	-720
IS-10	Quiquet et al. (2018) +10%	-750	-87	-714
GS	Quiquet et al. (2018)	-126	-85	-134
GS-5	Quiquet et al. (2018) +5%	-116	-89	-473
GS-10	Quiquet et al. (2018) +10%	-152	-103	-754
IS-EDC	EDC (Jouzel et al. 2007)	-167	-110	-487
IS-5-EDC	EDC +5%	-750	-109	-731
IS-10-EDC	EDC +10%	-757	-105	-731
GS-EDC	EDC	-746	-109	-703
GS-5-EDC	EDC +5%	-755	-107	-746
GS-10-EDC	EDC +10%	-757	-105	-722
IS-LR04	LR04 (Lisiecki and Raymo 2005)	-755	-93	-709
IS-5-LR04	LR04 +5%	-769	-37	-738
IS-10-LR04	LR04 +10%	-758	+539	-721
GS-LR04	LR04	-771	-96	-735
GS-5-LR04	LR04 +5%	-765	-64	-712
GS-10-LR04	LR04 +10%	-760	-326	-714

Table 3R: Elevation changes modelled for Talos Dome during interglacial MIS 5.5, 7.5, and 9.3 for sensitivity tests performed with the GRISLI ice sheet model, varying the Antarctic ice sheet initial conditions and the SO temperature forcing. Elevation variations at Talos Dome are calculated for interglacial time intervals when the simulated GRISLI ice thickness variations are maximized. Here we compare IS and GS simulations forced with the oceanic conditions derived from (i) Quiquet et al. (2018), (ii) the EDC δD record (Jouzel et al. 2007) and (iii) the LR04 stack (Lisiecki and Raymo 2005).

Figure 5R: Ice thickness variations at Talos Dome simulated with GRISLI starting from Antarctic ice sheet interglacial state (IS) and glacial state (GS). Here we compare the GRISLI simulations results obtained starting with Antarctic Interglacial conditions applying (a) the oceanic NADW forcing (Quiquet et al. 2018), (b) the LR04 oceanic forcing (Lisiecki and Raymo 2005), (c) the EDC forcing (Jouzel et al. 2007), and simulations starting at Antarctic Glacial conditions applying the (d) the oceanic NADW forcing (Quiquet et al. 2018), (e) the LR04 oceanic forcing (Lisiecki and Raymo 2005) and (f) the EDC forcing (Jouzel et al. 2007). We applied the original oceanic forcing - either NADW, LR04, or EDC - (blue curve), and the original forcing increased by 5% (yellow) and 10% (red).

Figure 6R: Wilkes Basin ice volume evolution (km^3) simulated with GRISLI starting from Antarctic ice sheet interglacial state (IS) and glacial state (GS). Here we show the GRISLI simulations results obtained from Antarctic ice sheet interglacial initial state and applying (a) the oceanic NADW forcing (Quiquet et al. 2018), (b) the LR04 forcing, and (c) the EDC forcing (Jouzel et al. 2007), and the GRISLI simulations starting at Antarctic glacial initial conditions and applying (d) the oceanic NADW forcing (Quiquet et al. 2018), (e) the LR04 forcing, and (f) the EDC forcing (Jouzel et al. 2007). We apply the original oceanic forcing – either NADW, LR04, or EDC – (blue curve), and the original forcing increased by 5% (yellow) and 10% (red).

Minor additional comments

- It may benefit the reader if the authors add a sentence to clarifying the sign of the change in $\delta^{18}\text{O}$ for a given elevation? Or edit the figure 2 to aid in understanding heavier isotopes indicates lower elevations. The only place I could find assigning a sign to the generic “elevation changes” is at line 180.

ANSWER 10

We agree with the reviewer that some sentences in the section “Interglacial elevation changes at Talos Dome from $\delta^{18}\text{O}$ records” are not clear in explaining that Talos Dome site was affected by elevation reduction during the interglacial periods. In order to avoid any kind of confusion and to increase the clarity of the text we performed the following changes:

- Line 177: Changed the section title from “Interglacial elevation changes at Talos Dome from $\delta^{18}\text{O}$ records” to “Interglacial elevation decrease at Talos Dome from $\delta^{18}\text{O}$ records”
- Lines 194-195: the specific anomalies are included in the text, specifying the sign “anomalous increase of the isotopic signal ($\Delta\delta^{18}\text{O}$) (see Methods) for MIS 5.5 (+1.68‰), MIS 7.5 (+0.68‰), and MIS 9.3 (+1.42‰)”
- Lines 195-197: a sentence is added to highlight the connection between the anomalous increase in the isotopic signal and the reduction in site elevation: “The less negative isotopic values during the late stages of the interglacials are connected to an increase in temperature at Talos Dome caused by elevation reduction at the site”
- Lines 200-202: the negative sign is added before the elevation decrease to avoid any misunderstanding “-100 to -200 m” and “elevation reduction (~ -300 m)” and general terms as “changes” and “variations” are substituted by specific language as “decrease” and “reduction” at lines 204 and 207.

- The resolution of figures 2&3 are too low.

ANSWER 10

We increased the figures resolution up to 300 dpi following the reviewer’s advice.

Reviewer #2 (Remarks to the Author):

Changes in response to **Reviewer #2** comments are highlighted in yellow in the tracked-record version of the manuscript.

This paper brings new insight into the retreats of the Antarctic ice sheet in the area of the Wilkes Subglacial Basin during interglacials back to MIS-9.3. The hypothesis testing of the ice core d180 and dD record signal during these interglacials is clearly laid out. I like that the paper combines ice sheet modeling with the ice elevation information from ice core d180 and dD records, this is quite convincing. The integration with the offshore evidence from IODP Site U1361 is also sound, and supports the main conclusion of the paper, that there was ice retreat of the ice margin in the Wilkes Basin area during MIS 9.3 and MIS 5.5. I have some comments, which include small- and medium-scale changes, after which, in my view, the manuscript will be ready to publish.

Changes:

The paper that describes the dating of the TALDICE ice core, Crotti et al., 2021, ref 50, should be cited much earlier on in the manuscript. There should be text in the introduction to explain how the current manuscript builds on the lead author’s 2021 paper, which also includes TALDICE dD results.

ANSWER 11

We agree with the reviewer that too little introduction is provided regarding the paper of Crotti et al. (2021), and that the addition of this information provides a clearer overview on the TALDICE ice core.

We have added the following sentences between lines 73 and 79 in order to introduce the study on which this manuscript is based: “The Talos Dome Ice Core (TALDICE) (159°11' E, 72°49'S, 2315 m a.s.l), which was drilled in a peripheral area of the East Antarctic Plateau (Figure 1), has been expected to be sensitive to grounding line retreat in the Wilkes Subglacial Basin since the beginning of the project (Frezzotti et al. 2004; Sutter et al. 2020). The recent extension of the TALDICE ice core chronology back to 343 ka (TALDICE-deep1) and the observation of a unique behaviour of the δD during the interglacial periods (Crotti et al. 2021) may therefore come in helpful for in deciphering the late Pleistocene dynamics of the peripheral EAIS”.

Line 82. Replace: “with late Pleistocene sedimentological and geochemical records from the marine core U1361A offshore Wilkes Subglacial Basin7 (Figure 1)” with “with late Pleistocene sedimentological and geochemical records from International Ocean Discovery Program (IODP) Hole U1361A, offshore of the Wilkes Subglacial Basin7 (Figure 1)”

ANSWER 12

We replaced the sentence as requested by the reviewer in lines 89 and 90.

Figure 1. IODP Site U1361 is misplaced. Currently marked on the continental shelf, it should be further north on the continental rise.

ANSWER 13

We thank the reviewer for this observation regarding the IODP misplacement in Figure 1. We have modified the Figure 1 including the IODP correct position.

Line 342 – Figure 3f, not 4f.

ANSWER 14

The number of the figure has been modified and correctly replaced with “3f” at line 389.

Line 349 – “ice sheet collapse” during MIS 11 – better to specify what is meant here – i.e. not the whole ice sheet collapsing but probably an ice margin retreat further into the Wilkes Basin than during subsequent interglacials. (In fact I’d speculate that the absence of clear MIS-11 ice at TALDICE might be seen as evidence of ice retreat a good ways back into the basin.)

ANSWER 15

We agree with the reviewer that the expression “ice sheet collapse” is too strong and not scientifically correct. We replaced it with the more specific sentence “retreat of about 700 km inland (from the present day position)” at lines 394-395.

Regarding the absence of MIS 11 in the isotopic signal at TALDICE, we agree that this is an interesting suggestion that reflects significant ice retreat. However, it could also be connected to mixing/folding processes acting in the portion of the core below 1548 m depth, as presented in Crotti et al. (2021). The three deep TALDICE layers dated with the ^{81}Kr technique reveal that the lowermost portion of the core is characterized by a mean age of about 400 ka, suggesting that during MIS 11 the Talos Dome site might have been covered by ice. However, this ice could have originated in neighbouring areas and then been transported to the Talos Dome site later on during the ice sheet expansion of MIS 10. Due to the high uncertainty connected to this speculation, we prefer to avoid proposing such a hypothesis about an ice-free Talos Dome area during MIS 11.

Add a figure to the supplement to illustrate the updated time scale for U1361. Perhaps including the original time scale and the new correlation to EDC AICC 2012.

ANSWER 16

We agree with the reviewer that a figure depicting the updated time scale for the U1361A sediment core can be useful for the readers, so we added it in the Supplementary Information.

We add the following notes in the manuscript to indicate the presence of the Figure S1 depicting the U1361A Ba/Al record on the sediment core original age scale (Wilson et al. 2018) and on the refined age scale used for this study based on the EDC AICC2012 chronology (Bazin et al. 2013).

- Line 335: added the reference to Supplementary information;
- Lines 527-529: The U1361A Ba/Al record on its original chronology and on the refined age is presented in Supplementary Information (Supplementary Figure 1- Figure 7R here).

The U1361A refined age scale (Figure 7R here) is presented in Figure 1 in the Supplementary Information.

Figure 7R: Refined age model for the U1361A sediment core on AICC2012 age scale. The U1361A Ba/Al record is shown on its original age scale (light blue curve with diamonds from Wilson et al. 2018) and on the AICC2012 age scale (blue curve with crosses). The age scale transfer is performed through the alignment of the Ba/Al record with the EDC δD profile (grey curve) on the AICC2012 age scale (Bazin et al. 2013). Tie points are represented by red dots.

Figure 3 shows the position of the retreated grounding line in the ice sheet model results, and Fig 8 the overall ice volume change. I'd like to see a more detailed map of where the ice is drawn down, e.g. with contour lines. This would be informative to include. Probably as an extra figure in the supporting information.

ANSWER 17

Following the reviewer’s advice, we produced the figure below (Figure 8R), in which the areas of ice thickness variation in the Wilkes Subglacial Basin are well highlighted for MIS 5.5 and MIS 9.3, according to the selected GS-5 simulation. We added this figure to the Supplementary Information (Supplementary Figure 5).

Figure 8R: Illustration of ice thickness variations in the vicinity of the Wilkes Subglacial Basin during (a) MIS 5.5 (115-128 ka), and (b) MIS 9.3 (318-332 ka), according to the GRISLI GS-5 simulation. The grounding line position at those different times is shown with green and red lines.

Notes to mention in the discussion, and perhaps follow up with in follow-up work: Neither of the two sets of models in the manuscript, which start in a glacial or interglacial state at 400 ka, MIS-11, represent the likely state of the ice sheet at that time, when the ice sheet was probably more retreated than at present. I would be interested to see the equivalent models for this starting condition. Still, the authors present the existing models clearly, and such further modeling can be future work.

ANSWER 18

We understand the comment of the reviewer and we agree that providing “more deglaciated” initial conditions with respect to the previously defined IS Antarctic ice sheet initial conditions could be an interesting test to perform. This simulation would represent a sensitivity test since the Antarctic Ice Sheet conditions during MIS 11 are still unknown, although some studies suggest that the AIS at 400 ka was probably more retreated in comparison to the present day (Blackburn et al. 2020).

Here we provide 3 additional sets of simulations with sensitivity tests in which the initial conditions of the Antarctic Ice Sheet at 400 ka are more deglaciated (Deglaciated State – DS) than the IS state presented in the original manuscript. To obtain this new initial state we perform a 10 ka simulation under present-day climate forcing except that we double the sub-

shelf melt starting from the IS Antarctic ice sheet state. The results of the IS, GS and DS experiments are presented here and discussed as suggested by the reviewer. **Error! Bookmark not defined.**

In particular, we focus on:

- i. Ice thickness variation at Talos Dome (Figure 9R and Table 4R)
- ii. Wilkes Subglacial Basin ice volume changes (Figure 10R).

The results of the sensitivity tests indicate that using deglaciated (DS) state as an initial condition at 400 ka only produces minor changes of the ice thickness at Talos Dome during MIS 9.3 in comparison to the IS simulations. In addition, after about 250 ka from the simulation beginning, the initial state is not relevant any more since the three experiments (DS, IS and GS) all produce a similar temporal evolution. There is only one exception with IS+10% being the only simulation producing a large retreat in the Wilkes Basin during MIS 5.5 while GS+10% and DS+10% simulate a very limited retreat. This illustrates that the grounding line in this sector is very sensitive to long-term feedbacks such as glacial isostasy and internal temperatures.

For all the experiments, considering a given oceanic forcing scenario, we simulate a larger retreat during MIS 9.3 compared to MIS 5.5. This can be counter-intuitive given that the oceanic forcing is slightly stronger during MIS 5.5 than during MIS 9.3 (Fig. 4R). However, the larger retreat during MIS 9.3 might be connected to the fact that (i) the glacial period that precedes MIS 9.3 (MIS 10) is weaker than the one that precedes MIS 5.5, or (ii) to the likely reduced ice sheet coverage of the Wilkes Subglacial Basin during MIS 11. Both hypotheses might explain the larger retreat during MIS 9.3 compared to MIS 5.5.

We conclude that the DS test is a useful sensitivity test demonstrating that strong variations in elevation can easily be simulated over MIS 9.3 at TALDICE. It also shows that a better knowledge of the Antarctic Ice Sheet conditions during MIS 11 is necessary for more realistic simulations, especially for MIS 9.3. It confirms that the results of our modelling simulations should be taken as indications for the possibility of strong ice sheet loss in the Wilkes Basin region during MIS 9.3, but constraints are still missing to provide a more quantitative interpretation with modelling outputs.

Overall, as rightly noted by the reviewer, the initial “Deglaciated State” represents more realistic Antarctic ice sheet conditions at 400 ka in comparison to the Interglacial State. For this reason, we decided to display the DS simulations in the revised version of the manuscript, substituting these for the previous IS simulations. The DS state is introduced between lines 243-252 of the manuscript as following: “The GRISLI experiments labelled DS (Deglaciated Start) adopt an initial Antarctic interglacial ice sheet state, while the GS (Glacial Start) experiments use a Last Glacial Maximum (21 ka) initial state(Quiquet et al. 2018). The initial ice sheet GS state is simulated as in Quiquet et al.(Quiquet et al. 2018), while the DS state has been defined for this study and is obtained after 10 ka simulation under present-day climate forcing (1976-2016) and doubling the original sub-shelf melting(Quiquet et al. 2018), starting from the Interglacial Start (IS) represented by the present-day Antarctic ice sheet state. The DS state is considered more likely to reproduce Antarctic ice sheet conditions during MIS 11 (~400 ka) in comparison to the IS state. Simulations have also been performed applying an IS

state and those results are presented in the Supplementary Table 1 and Supplementary Figure 3 and 4”.

In particular, we replace the elevation changes at Talos Dome calculated from the IS simulations in Table 1 (in the manuscript) with the elevation variations computed by the DS simulations and presented here in Table 4R. We substitute the IS acronym with the DS acronym in the section “Sensitivity tests with the GRISLI ice sheet model”. We modify the original text between lines 293 and 297 as following: “A ~700 m elevation reduction at Talos Dome is simulated over MIS 9.3 and a significant grounding line retreat of the Wilkes Subglacial Basin ice sheet is modelled for all the DS experiments (Table 1, Supplementary Table 1). On the other hand, the DS simulations depict a less dynamic Wilkes Subglacial Basin ice sheet for MIS 5.5”. The modelled Wilkes Subglacial Basin Ice Sheet volume variations and Talos Dome ice thickness changes over the past 350 ka are shown for GS, IS and DS simulations in Figures 9R and 10R.

Figure 9R: Ice thickness variations at Talos Dome simulated with GRISLI starting from Antarctic ice sheet interglacial state (IS), glacial state (GS) and deglaciaded state (DS). Here we compare the GRISLI simulations results obtained applying the oceanic NADW forcing (Quiquet et al. 2018) starting from Antarctic interglacial (a), glacial (b) and deglaciaded initial conditions (c). We applied the original oceanic forcing (blue curve), and the original forcing increased by 5% (yellow) and 10% (red).

Figure 10R: Wilkes Basin Volume variations (km^3) simulated with GRISLI starting from Antarctic ice sheet interglacial state (IS), glacial state (GS) and deglaciaded state (DS). Here we compare the GRISLI simulations results obtained applying the oceanic NADW forcing (Quiquet et al. 2018) starting from Antarctic interglacial (a), glacial (b) and deglaciaded initial conditions (c). We applied the original oceanic forcing (blue curve), and the original forcing increased by 5% (yellow) and 10% (red).

Here we also present an additional set of 6 simulations to provide a complete record of sensitivity tests. In particular, we perform GRISLI simulations adopting the DS initial state for the Antarctic ice sheet and forcing the model with the EDC and LR04 oceanic indexes presented in ANSWER 10 as suggested by Reviewer #1. The computed ice thickness variations at Talos Dome are displayed in Table 4R. The ice thickness changes at Talos Dome and the Wilkes Subglacial Basin ice volume changes over time forced with the EDC and LR04 oceanic records are plotted in Figures 11R and 12R and compared with the simulations presented in Answer 10. Figures 11R and 12R are displayed in the Supplementary Information as Figures S3 and S4. As stated in Answer 10, also concerning the DS simulation, forcing with the EDC and LR04 oceanic records produces greater retreat during MIS 5.5 and MIS 9.3 in comparison to the simulations forced by the ODP 980 record. The new experiments produce a strong ice thinning at Talos Dome and large ice volume decrease in the Wilkes Subglacial Basin that is irreconcilable with our data. For this reason, we display the DS simulations forced with EDC and LR04 oceanic indexes only in the Supplementary Information.

Ocean forcing		GRISLI elevation changes at Talos Dome (m)		
Time interval for max elevation anomaly at TALDICE (ka)		115-128	233-241	321-332
IS	Quiquet et al. (2018)	-132	-103	-749
IS-5	Quiquet et al. (2018) +5%	-126	-101	-720
IS-10	Quiquet et al. (2018) +10%	-750	-87	-714
GS	Quiquet et al. (2018)	-126	-85	-134
GS-5	Quiquet et al. (2018) +5%	-116	-89	-473
GS-10	Quiquet et al. (2018) +10%	-152	-103	-754
DS	Quiquet et al. (2018)	-123	-103	-757
DS-5	Quiquet et al. (2018) +5%	-114	-104	-754
DS-10	Quiquet et al. (2018) +10%	-146	-105	-749
DS-EDC	EDC (Jouzel et al. 2007)	-758	-107	-742
DS-5-EDC	EDC +5%	-754	-108	-744
DS-10-EDC	EDC +10%	-760	-110	-751
DS-LR04	LR04 (Lisiecki and Raymo 2005)	-766	-95	-736
DS-5-LR04	LR04 +5%	-766	-604	-737
DS-10-LR04	LR04 +10%	-765	334	-723

Table 4R: Elevation changes modelled for Talos Dome during interglacial MIS 5.5, 7.5, and 9.3. Elevation changes are modelled for sensitivity tests performed with the GRISLI ice sheet model, varying the Antarctic ice sheet initial conditions and the SO temperature forcing. Elevation variations at Talos Dome are calculated for interglacial time intervals when the simulated GRISLI ice thickness variations are maximized. Here we compare IS, GS, and DS simulations forced with the oceanic conditions from Quiquet et al. (2018), as well as using EDC and LR04 records for ocean forcing.

Figure 11R: Ice thickness variations at Talos Dome from GRISLI Interglacial State (a-c), Glacial State (d-f) and Deglaciating State (g-i) simulations during the past 350 ka. We applied the original oceanic forcing from Quiquet et al. (2018) (a, d, g), the oceanic forcing derived from the stacked benthic record LR04 (Lisiecki and Raymo 2005) and the oceanic forcing computed from the EDC δD record (Jouzel et al. 2007). The simulations are forced with the original oceanic index (blue curve), with the forcing increased by 5% (yellow curve) and 10% (red curve).

Figure 12R: Wilkes Subglacial Basin ice volume evolution from GRISLI Interglacial State (a-c), Glacial State (d-f) and Deglaciating State (g-i) simulations during the past 350 ka. We applied the original oceanic forcing from Quiquet et al. (2018) (a, d, g), the oceanic forcing derived from the stacked benthic record LR04 (Lisiecki and Raymo 2005) and the oceanic forcing computed from the EDC δD record (Jouzel et al. 2007). The simulations are forced with the original oceanic index (blue curve), with the forcing increased by 5% (yellow curve) and 10% (red curve).

The double peaks in $d18O$ during MIS 9.3 and 5.5 are described in the results and compared to recent sea surface temperature results from the South Indian Ocean (Fig 3f). I think the authors can go further with their observations here. In both cases the second peak is about ~ 10 kyr after the first, and close to the time of peak southern hemisphere insolation. This hints that local insolation is a factor in Antarctic ice margin stability during these interglacials. It has been hypothesized for a while now that Antarctic ice responded to local insolation, as a way to explain the 40 kyr cyclicity from ~ 3 to 1 Ma (e.g. Raymo, Lisiecki, and Nisancioglu, 2006). Anyway, it is very interesting to find suggestions of local insolation control in the last 1 Myr, it might serve as a hypothesis for a future paper.

ANSWER 19

We thank the reviewer for this comment and for this additional input to our manuscript. To incorporate the reviewer's advice in the manuscript we added the following statement at lines 378-383, including the reference of Raymo et al. (2006) and a reference to the recent work of Wu et al. (2021) on the Lambert Glacier-Amery Ice Shelf system: "The distinctive late interglacial $\delta^{18}\text{O}$ peaks in the TALDICE record during MIS 5.5 and MIS 9.3 occur approximately 5-7 ka after the interglacial isotopic maxima and are synchronous with the local summer insolation maxima at 65°S (Figure 3a), which supports the hypothesis that local insolation could play a role, together with ocean temperature and upwelling, in modulating the ice sheet margin dynamics in the Wilkes Subglacial Basin (Raymo et al. 2006; Wu et al. 2021)".

References

- Ai, Xuyuan E., Anja S. Studer, Daniel M. Sigman, Alfredo Martínez-García, François Fripiat, Lena M. Thöle, Elisabeth Michel, Julia Gottschalk, Laura Arnold, Simone Moretti, Mareike Schmitt, Sergey Oleynik, Samuel L. Jaccard, and Gerald H. Haug. 2020. "Southern Ocean Upwelling, Earth's Obliquity, and Glacial-Interglacial Atmospheric CO₂ Change." *Science* 370(6522):1348–52.
- Asay-Davis, Xylar S., Nicolas C. Jourdain, and Yoshihiro Nakayama. 2017. "Developments in Simulating and Parameterizing Interactions Between the Southern Ocean and the Antarctic Ice Sheet." *Current Climate Change Reports* 3(4):316–29.
- Bazin, L., A. Landais, B. Lemieux-Dudon, H. Toyé Mahamadou Kele, D. Veres, F. Parrenin, P. Martinerie, C. Ritz, E. Capron, V. Lipenkov, M. F. Loutre, D. Raynaud, B. Vinther, A. Svensson, S. O. Rasmussen, M. Severi, T. Blunier, M. Leuenberger, H. Fischer, V. Masson-Delmotte, J. Chappellaz, and E. Wolff. 2013. "An Optimized Multi-Proxy, Multi-Site Antarctic Ice and Gas Orbital Chronology (AICC2012): 120-800 Ka." *Climate of the Past* 9(4):1715–31.
- Blackburn, T., G. H. Edwards, S. Tulaczyk, M. Scudder, G. Piccione, B. Hallet, N. McLean, J. C. Zachos, B. Cheney, and J. T. Babbe. 2020. "Ice Retreat in Wilkes Basin of East Antarctica during a Warm Interglacial." *Nature* 583(7817):554–59.
- Blasco, Javier, Ilaria Tabone, Jorge Alvarez-Solas, Alexander Robinson, and Marisa Montoya. 2019. "The Antarctic Ice Sheet Response to Glacial Millennial-Scale Variability." *Climate of the Past* 15(1):121–33.
- Buizert, C., Michael Sigl, Mirko Severi, Bradley R. Markle, Justin J. Wettstein, Joseph R. McConnell, Joel B. Pedro, Harald Sodemann, Kumiko Goto-Azuma, Kenji Kawamura, Shuji Fujita, Hideaki Motoyama, Motohiro Hirabayashi, Ryu Uemura, Barbara Stenni, Frédéric Parrenin, Feng He, T. J. Fudge, and Eric J. Steig. 2018. "Abrupt Ice-Age Shifts in Southern Westerly Winds and Antarctic Climate Forced from the North." *Nature* 563(7733):681–85.
- Chadwick, M., C. S. Allen, L. C. Sime, and C. D. Hillenbrand. 2020. "Analysing the Timing of Peak Warming and Minimum Winter Sea-Ice Extent in the Southern Ocean during MIS 5e." *Quaternary Science Reviews* 229:106134.
- Crotti, Ilaria, Amaelle Landais, Barbara Stenni, Lucie Bazin, Massimo Frezzotti, Florian Ritterbusch, Zheng-tian Lu, Wei Jiang, Guo-min Yang, Anais Orsi, Roxanne Jacob, Elise

- Fourr, Giuliano Dreossi, and Carlo Barbante. 2021. "An Extension of the TALDICE Ice Core Age Scale Reaching Back to MIS 10.1." *Quaternary Science Reviews* 266(107078).
- Ferrari, Raffaele, Malte F. Jansen, Jess F. Adkins, Andrea Burke, Andrew L. Stewart, and Andrew F. Thompson. 2014. "Antarctic Sea Ice Control on Ocean Circulation in Present and Glacial Climates." *Proceedings of the National Academy of Sciences of the United States of America* 111(24):8753–58.
- Fogwill, Christopher J., Christian S. M. Turney, Katrin J. Meissner, Nicholas R. Golledge, Paul Spence, Jason L. Roberts, Mathew H. England, Richard T. Jones, and Lionel Carter. 2014. "Testing the Sensitivity of the East Antarctic Ice Sheet to Southern Ocean Dynamics: Past Changes and Future Implications." *Journal of Quaternary Science* 29(1):91–98.
- Frezzotti, Massimo, Gabriele Bitelli, Paola De Michelis, Alberto Deponti, Alessandro Forieri, Stefano Gandolfi, Valter Maggi, Francesco Mancini, Frédérique Remy, Ignazio E. Tabacco, Stefano Urbini, Luca Vittuari, and Achille Zirizzotti. 2004. "Geophysical Survey at Talos Dome, East Antarctica: The Search for a New Deep-Drilling Site." *Annals of Glaciology* 39(2002):423–32.
- Golledge, N. R., L. Menviel, L. Carter, C. J. Fogwill, M. H. England, G. Cortese, and R. H. Levy. 2014. "Antarctic Contribution to Meltwater Pulse 1A from Reduced Southern Ocean Overturning." *Nature Communications* 5:1–10.
- Jouzel, J., V. Masson-Delmotte, O. Cattani, G. Dreyfus, S. Falourd, G. Hoffmann, B. Minster, J. Nouet, J. M. Barnola, J. Chappellaz, H. Fischer, J. C. Gallet, S. Johnsen, M. Leuenberger, L. Loulergue, D. Luethi, H. Oerter, F. Parrenin, G. Raisbeck, D. Raynaud, A. Schilt, J. Schwander, E. Selmo, R. Souchez, R. Spahni, B. Stauffer, J. P. Steffensen, B. Stenni, T. F. Stocker, J. L. Tison, M. Werner, and E. W. Wolff. 2007. "Orbital and Millennial Antarctic Climate Variability over the Past 800,000 Years." *Science* 317(5839):793–96.
- Landais, A., B. Stenni, V. Masson-Delmotte, J. Jouzel, A. Cauquoin, E. Fourre, B. Minster, E. Selmo, T. Extier, M. Werner, F. Vimeux, R. Uemura, I. Crotti, and A. Grisart. 2021. "Interglacial Antarctic-Southern Ocean Climate Decoupling Due to Moisture Source Area Shifts." *Nature Geoscience* 14:918–23.
- Landais, A., B. Stenni, V. Masson Demotte, J. Jouzel, A. Cauquoin, E. Fourré, B. Minster, E. Selmo, T. Extier, M. Werner, F. Vimeux, R. Uemura, I. Crotti, and A. Grisart. 2021. "Interglacial Antarctic–Southern Ocean Climate Decoupling Due to Moisture Source Area Shifts." *Nature Geoscience* 14:918–23.
- Lisiecki, L. E., and M. E. Raymo. 2015. "The Holocene and MIS 1 : Not Quite the Same Holocene Pleistocene Thousand of Years BP." 1(Mis 1):2015.
- Lisiecki, Lorraine E., and Maureen E. Raymo. 2005. "A Pliocene–Pleistocene Stack of 57 Globally Distributed Benthic $\delta^{18}\text{O}$ Records." *Paleoceanography* 20(1):1–17.
- Masson Delmotte, V., D. Buiron, A. Ekaykin, M. Frezzotti, H. Gallée, J. Jouzel, G. Krinner, A. Landais, H. Motoyama, H. Oerter, K. Pol, D. Pollard, C. Ritz, E. Schlosser, L. C. Sime, H. Sodemann, B. Stenni, R. Uemura, and F. Vimeux. 2011. "A Comparison of the Present and Last Interglacial Periods in Six Antarctic Ice Cores." *Climate of the Past* 7(2):397–423.
- Parrenin, F., L. Bazin, E. Capron, A. Landais, B. Lemieux-Dudon, and V. Masson-Delmotte.

2015. “IceChrono1: A Probabilistic Model to Compute a Common and Optimal Chronology for Several Ice Cores.” *Geoscientific Model Development* 8(5):1473–92.
- Quiquet, Aurélien, Christophe Dumas, Catherine Ritz, Vincent Peyaud, and Didier M. Roche. 2018. “The GRISLI Ice Sheet Model (Version 2 . 0): Calibration and Validation for Multi-Millennial Changes of the Antarctic Ice Sheet.” *Geoscientific Model Development* 11:5003–25.
- Raymo, M. E., L. E. Lisiecki, Kerim H. Nisancioglu, Southern Hemispheres, and O. Instead. 2006. “Plio-Pleistocene Ice Volume, Antarctic Climate, and the Global D18O Record.” *Nature* 313(July):492–95.
- Seroussi, H el ene, Sophie Nowicki, Antony J. Payne, Heiko Goelzer, William H. Lipscomb, Ayako Abe-Ouchi, C ecile Agosta, Torsten Albrecht, Xylar Asay-Davis, Alice Barthel, Reinhard Calov, Richard Cullather, Christophe Dumas, Benjamin K. Galton-Fenzi, Rupert Gladstone, Nicholas R. Golledge, Jonathan M. Gregory, Ralf Greve, Tore Hattermann, Matthew J. Hoffman, Angelika Humbert, Philippe Huybrechts, Nicolas C. Jourdain, Thomas Kleiner, Eric Larour, Gunter R. Leguy, Daniel P. Lowry, Christopher M. Little, Mathieu Morlighem, Frank Pattyn, Tyler Pelle, Stephen F. Price, Aur elien Quiquet, Ronja Reese, Nicole Jeanne Schlegel, Andrew Shepherd, Erika Simon, Robin S. Smith, Fiammetta Straneo, Sainan Sun, Luke D. Trusel, Jonas Van Breedam, Roderik S. W. Van De Wal, Ricarda Winkelmann, Chen Zhao, Tong Zhang, and Thomas Zwinger. 2020. “ISMIP6 Antarctica: A Multi-Model Ensemble of the Antarctic Ice Sheet Evolution over the 21st Century.” *Cryosphere* 14(9):3033–70.
- Shukla, S. K., X. Crosta, and M. Ikehara. 2021. “Sea Surface Temperatures in the Indian Sub-Antarctic Southern Ocean for the Last Four Interglacial Periods.” *Geophysical Research Letters* 48(8):1–11.
- Sutter, J., O. Eisen, M. Werner, K. Grosfeld, T. Kleiner, and H. Fischer. 2020. “Limited Retreat of the Wilkes Basin Ice Sheet During the Last Interglacial.” *Geophysical Research Letters* 47(13).
- Uemura, Ryu, Hideaki Motoyama, Val erie Masson-Delmotte, Jean Jouzel, Kenji Kawamura, Kumiko Goto-Azuma, Shuji Fujita, Takayuki Kuramoto, Motohiro Hirabayashi, Takayuki Miyake, Hiroshi Ohno, Koji Fujita, Ayako Abe-Ouchi, Yoshinori Iizuka, Shinichiro Horikawa, Makoto Igarashi, Keisuke Suzuki, Toshitaka Suzuki, and Yoshiyuki Fujii. 2018. “Asynchrony between Antarctic Temperature and CO 2 Associated with Obliquity over the Past 720,000 Years.” *Nature Communications* 9(1):1–11.
- Vimeux, F., V. Masson, G. Delaygue, J. Jouzel, J. R. Petit, and M. Stievenard. 2001. “A 420,000 Year Deuterium Excess Record from East Antarctica: Information on Past Changes in the Origin of Precipitation at Vostok.” *Journal of Geophysical Research Atmospheres* 106(D23):31863–73.
- Waelbroeck, C., L. Labeyrie, E. Michel, J. C. Duplessy, and J. F. Mcmanus. 2002. “Sea-Level and Deep Water Temperature Changes Derived from Benthic Foraminifera Isotopic Records.” *Quaternary Science Reviews* 21:295–305.
- Wilson, David J., Rachel A. Bertram, Emma F. Needham, Tina van de Flierdt, Kevin J. Welsh, Robert M. McKay, Anannya Mazumder, Christina R. Riesselman, Francisco J. Jimenez-Espejo, and Carlota Escutia. 2018. “Ice Loss from the East Antarctic Ice Sheet during Late Pleistocene Interglacials.” *Nature* 561(7723):383–86.

Wolff, E. W., H. Fischer, and R. Röthlisberger. 2009. "Glacial Terminations as Southern Warmings without Northern Control." *Nature Geoscience* 2(3):1–4.

Wu, Li, David J. Wilson, Rujian Wang, Sandra Passchier, Wout Krijgsman, Xun Yu, Tingyu Wen, Wenshen Xiao, and Zhifei Liu. 2021. "Late Quaternary Dynamics of the Lambert Glacier-Amery Ice Shelf System, East Antarctica." *Quaternary Science Reviews* 252.

Reviewers' Comments:

Reviewer #1:

Remarks to the Author:

This is my second review (reviewer 1) of the Crotti and others manuscript. I'm impressed by this group's thoughtful responses to my suggestions and questions. I particularly enjoyed reading the result of their principal component analysis, and I believe it has provided the team with a quantitative framework to argue that the d18O timeseries at talos dome are distinct. I fully support the publication of the manuscript, I only want to revisit one of my original points that seemed to not be fully appreciated by the authors. The authors often refer to EDC as an atmospheric record—and it is, strictly speaking a measurement of Antarctica's surface temperatures. The point I was trying to make by including mention of Anderson and others (2021) or Blasco et al (2019) is that the atmospheric temperature is controlled by SO temperatures. I understand the authors point that SST is not the exact temperature at the grounding lines but I'm not sure how much that would matter for the purposes of driving an ice sheet model. Any modeled ice sheet response to a given change in ocean temperature is controlled by the heat exchange coefficient between the ocean and ice. This coefficient is unknown but could be calibrated with the kind of geologic record that these authors have produced. My point was, why not use the high-resolution EDC, knowing that the temperatures in no way represent ocean temperatures at the grounding line, but that the secular variations are reliable and higher resolution and then calibrate the heat exchange coefficient to match the observed d18O response at Talos dome. Just a minor point. I hope my comments helped, Terry Blackburn

Reviewer #2:

Remarks to the Author:

The authors have addressed my comments and those of the other reviewer very thoroughly, it is a very nice paper. I have only one further quick change to request, in the last point, below:

- The Site U1361 refined age scale will be a valuable resource for anyone working on that IODP site, great that it has been added as Figure 1 in the introductory information, as well in the Pangaea database.
- Thank you for the additional modeling with the DS deglacial state starting condition (and for the models based on different forcing records). Interesting that the result is mostly insensitive to the starting condition, and that the results from EDC and LR04 forcing do not lead to consistency with the Talos Dome record. The new models make the paper more robust.
- Thank you for adding the contour maps of modeled ice sheet drawdown in interglacials 5.5 and 9.3. Great to see the locations where ice retreat would happen.
- The additional text about the double d18O peaks in MIS 5.5 and 9.3 at Talos Dome is good (Answer 19), but the age intervals between these double peaks are not reported correctly. At least they do not match the peak-to-peak d18O age intervals evident in Figures 3 and 5. The peaks are separated by ~9 ka in MIS 9.3 and by ~11 ka in MIS 5.5, not by 5-7 ka as in the revision.

ANSWER TO REVIEWERS (Review #2)

We thank the reviewers for their comments and suggestions that improved the scientific discussion and the quality of the manuscript. In particular, we thank them for their positive final response and for their support to our manuscript. Here we present our final answers to the reviewers' feedback.

In this version of the manuscript we also included the editor's suggestions and those changes are highlighted in **green**. In particular, we revised the authors' affiliations and included the editor's suggestion at line 23. Sections order has been changed in order to comply the editorial requests. We also modified the data availability statement as suggested by the data availability guidelines and the Acknowledgements section.

Reviewer #1 (Remarks to the Author):

Changes in response to **Reviewer #1** comments are highlighted in **light blue** in the tracked-record version of the manuscript.

This is my second review (reviewer 1) of the Crotti and others manuscript. I'm impressed by this group's thoughtful responses to my suggestions and questions. I particularly enjoyed reading the result of their principal component analysis, and I believe it has provided the team with a quantitative framework to argue that the d18O timeseries at talos dome are distinct. I fully support the publication of the manuscript, I only want to revisit one of my original points that seemed to not be fully appreciated by the authors. The authors often refer to EDC as an atmospheric record—and it is, strictly speaking a measurement of Antarctica's surface temperatures. The point I was trying to make by including mention of Anderson and others (2021) or Blasco et al (2019) is that the atmospheric temperature is controlled by SO temperatures. I understand the authors point that SST is not the exact temperature at the grounding lines but I'm not sure how much that would matter for the purposes of driving an ice sheet model. Any modeled ice sheet response to a given change in ocean temperature is controlled by the heat exchange coefficient between the ocean and ice. This coefficient is unknown but could be calibrated with the kind of geologic record that these authors have produced. My point was, why not use the high-resolution EDC, knowing that the temperatures in no way represent ocean temperatures at the grounding line, but that the secular variations are reliable and higher resolution and then calibrate the heat exchange coefficient to match the observed d18O response at Talos Dome. Just a minor point. I hope my comments helped, Terry Blackburn.

ANSWER 1

We thank the reviewer for his insightful suggestions and comments that expanded the analysis proposed in our first version of the manuscript and increased the robustness of the scientific message. In particular, we greatly appreciated the comments regarding the different records that could be used to force the SO sub-surface conditions in the GRISLI model. In fact, we agree with the reviewer that we will never have the temperature at the grounding line in the geological record; for such a reason we need to find a proxy for it. We do agree that we could also use EDC as a proxy for SO surface temperatures. However, in our first revision's comments we wanted to indicate that there is a potential decoupling between the surface temperature and the sub-surface temperature (which matters for sub-shelf melt). There could be some anti-phase relationship at times due, for example, to brines sinking link to sea ice formation (Colleoni et al., 2018). As such, although the amplitude of the signal at EDC does not matter much for sub-shelf melt parametrisation, the temporal evolution might not be the best option. Generally, it is a question of great importance for simulating the Antarctic ice sheet evolution and we thank the reviewer the essential suggestion to use various proxy for SO temperature to perform simulations. However, hard constraints on the ice sheet evolution are still lacking in the geological record and such experiments will only provide sensitivity tests for hypothetical scenarios.

Reviewer #2 (Remarks to the Author):

Changes in response to **Reviewer #2** comments are highlighted in yellow in the tracked-record version of the manuscript.

The authors have addressed my comments and those of the other reviewer very thoroughly, it is a very nice paper. I have only one further quick change to request, in the last point, below:

- The Site U1361 refined age scale will be a valuable resource for anyone working on that IODP site, great that it has been added as Figure 1 in the introductory information, as well in the Pangaea database.
- Thank you for the additional modeling with the DS deglacial state starting condition (and for the models based on different forcing records). Interesting that the result is mostly insensitive to the starting condition, and that the results from EDC and LR04 forcing do not lead to consistency with the Talos Dome record. The new models make the paper more robust.
- Thank you for adding the contour maps of modeled ice sheet drawdown in interglacials 5.5 and 9.3. Great to see the locations where ice retreat would happen.

ANSWER 2

We thank the reviewer #2 for his/her contribution and comments which helped us to correct inaccuracies and elaborate a more complete and robust analysis. We are especially thankful for the suggestion regarding the integration of the more deglacialized initial state of the Antarctic Ice Sheet which, with the sensitivity experiments suggested by reviewer #1, increase the strength of our study.

- The additional text about the double d18O peaks in MIS 5.5 and 9.3 at Talos Dome is good (Answer 19), but the age intervals between these double peaks are not reported correctly. At least they do not match the peak-to-peak d18O age intervals evident in Figures 3 and 5. The peaks are separated by ~9 ka in MIS 9.3 and by ~11 ka in MIS 5.5, not by 5-7 ka as in the revision.

ANSWER 3

We are thankful to the reviewer for this final comment and we agree that the age intervals are not reported correctly. As the reviewer points out correctly, the peaks are separated by ~11 ka in MIS 5.5 by ~9 ka in MIS 9.3, not by 5-7 ka as written in answer 19 and in the manuscript. We modified the manuscript at line 336 as following: "...peaks in the TALDICE record during MIS 5.5 and MIS 9.3 occurred approximately 9-11 ka".

References

Colleoni, F., De Santis, L., Montoli, E., Olivo, E., Sorlien, C. C., Bart, P. J., Gasson, E. G. W., Bergamasco, A., Sauli, C., Wardell, N., & Prato, S. (2018). Past continental shelf evolution increased Antarctic ice sheet sensitivity to climatic conditions. *Scientific Reports*, 8(1), 1–12. <https://doi.org/10.1038/s41598-018-29718-7>